



# Thermodynamic and dynamic drivers underlying extreme central Arctic sea ice loss

Zhenlin Li[1,2], Fei Huang[1,2,3], Jian Shi[1], Ruichang Ding[1], Shumeng Zhang[1]

[1]State Key Laboratory of Physical Oceanography/College of Oceanic and Atmospheric Sciences, Ocean University of China, Qingdao, 266000, China
[2]Academy of Future Ocean/Frontier Science Center for Deep Ocean Multispheres and Earth System, Ocean University of China, Qingdao, 266000, China
[3]Laboratory for Ocean Dynamics and Climate, Qingdao Marine Science and Technology Center, Qingdao, 266000, China

*Correspondence to*: Fei Huang (huangf@ouc.edu.cn) and Jian Shi (shijian@ouc.edu.cn)

**Abstract.** Variability of sea ice in the central Arctic is pivotal to the entire Arctic region, yet in-depth understanding of its characteristics and underlying mechanisms remains elusive. We investigate the characteristics and mechanisms of extremely low sea ice concentration (SIC) events (ELSEs) in the central Arctic by analysing the reanalysis data. First, we define the area where climatological SIC exceeds 90% as the central Arctic. Moreover, this paper primarily investigates the relative variations in sea ice, i.e., the standardized sea ice anomalies, which highlight common characteristics of sea ice variations. Based on the Empirical Orthogonal Function method, we identify the first two dominant modes of relative variations of ELSEs within the central Arctic: the East–West Seesaw Mode (EWSM) and the Pacific–Atlantic Seesaw Mode (PASM). By using the sea ice budget diagnostic method for comparison, it is found that the thermodynamic (dynamic) contributions of EWSM and PASM account for 68% (32%) and 72% (28%), respectively. Thermodynamically, both the EWSM and the PASM are dominated by local diabatic heating, with their maintenance and intensification facilitated by water vapor and cloud feedbacks. Dynamically, sea ice advection plays an important role in the formation of the two modes. In particular, the upper tropospheric divergence anomalies over the North Pacific induce Rossby wave train that modulate the EWSM. Overall, this study elucidates the key drivers of ELSEs in the central Arctic, which enriches our knowledge of the complex cryosphere processes.

## 1 Introduction

The Arctic region is profoundly affected by climate change and reciprocally exerts a pivotal influence on the global climate system (Grunseich and Wang, 2016; Kim et al., 2014; Meng et al., 2023; Wu, 2017), which is largely attributed to the intricate sea-ice-atmosphere coupling (Liang et al., 2022a, b; Tian et al., 2022). Amidst the context of global warming, the Arctic experiences a greater warming rate that is approximately twice the global average, a phenomenon commonly referred to as Arctic Amplification (Pithan and Mauritsen, 2014; Serreze and Barry, 2011). Recent research has highlighted that Arctic Amplification has been underestimated (Chylek et al., 2022; Rantanen et al., 2022). Both observations and modeling



studies have revealed that the Arctic region is warming at a rate approximately four times faster than the global average, with the potential to surpass eight times that of the global average in winter (Davy and Griewank, 2023). This rapid warming in the Arctic has led to a significant decline trend in Arctic sea ice (Cai et al., 2021a; Nghiem et al., 2007; Park et al., 2020; Roach and Blanchard-Wrigglesworth, 2022; Sumata et al., 2023; Yang and Magnusdottir, 2018; Yu et al., 2020; Yu and
Zhong, 2018).

In terms of the physical processes, the mean state of Arctic sea ice is predominantly influenced by the prevailing local atmospheric conditions within the Arctic region (Liang et al., 2022b). For example, atmospheric circulation acts as a pivotal modulator to drive the evolution of Arctic sea ice. It drives sea ice variations through thermodynamic processes (e.g., formation and melting) and dynamic processes (e.g., redistribution) (Wang et al., 2022). The atmospheric circulation in the
Arctic is intimately connected to the mid-latitudes via teleconnections (Baxter et al., 2019; Deng and Dai, 2024; Grunseich and Wang, 2016; Yu and Zhong, 2018). This linkage underscores the profound interplay between Arctic sea ice and global weather and climate, which is a cornerstone for the accuracy of weather forecasts, the understanding of extreme climatic events, and the progress of climate change research (Meng et al., 2023). What's more, there are numerous positive feedbacks between Arctic Amplification and diminishing Arctic sea ice (Wu et al., 2019), such as surface albedo feedback (Screen and
Simmonds, 2010; Taylor et al., 2013), cloud feedback (Vavrus, 2004), water vapor feedback (Graversen and Wang, 2009), temperature feedback (Pithan and Mauritsen, 2014), and river heat–sea ice feedback (Park et al., 2020). Recently, a multi-factor feedback involving the coupling of thermodynamics and dynamics has been observed (Voosen, 2020). This feedback encompasses elements of wind, ocean currents, ocean waves, sea ice deformation and fracture, and the presence of underwater warm blobs. The complex interplay of these positive feedbacks renders Arctic sea ice increasingly sensitive and
challenges its capacity for recovery.

Arctic sea ice exhibits pronounced seasonal fluctuations that peak in March and reach the minimum in September (Cavalieri and Parkinson, 2012; Onarheim et al., 2018). Consequently, the reduction of Arctic sea ice during the summer and autumn seasons, particularly the extremely low sea ice concentration (SIC) events (ELSEs), has gained widespread attention (Wang et al. 2009, 2022; Jeong et al. 2022; Li et al. 2022; Liang et al. 2022a; Moore et al. 2022; Roach and Blanchard-
Wrigglesworth 2022; Tian et al. 2022; Zhou et al. 2024). The occurrence timing of ice-free Arctic in summer remains a hotly debated topic (Kim et al., 2023; Topál and Ding, 2023; Zhou et al., 2022). Despite the smaller amplitude of sea ice decline during winter and spring, it is the critical period for the recovery of sea ice (Cornish et al., 2022). Moreover, Arctic sea ice reductions during winter and spring are known to trigger amplified reductions in summer sea ice through a series of positive feedbacks (Bi et al., 2023; Schröder et al., 2014; Smith et al., 2018). This amplification is not only scientifically intriguing
but also of paramount importance for predictive modeling (Mills and Walsh, 2014; Yang and Magnusdottir, 2018; Zhou et al., 2022).

Additionally, the reduction of Arctic sea ice is spatially heterogeneous, which is attributable to the spatial variation of thermodynamic and dynamic processes driven by atmospheric and oceanic circulation (Cai et al., 2021b; Spreen et al., 2020; Sumata et al., 2022; Wu and Ding, 2023). The trend of sea ice reduction is notably significant in the marginal seas along the



Eurasian and the North American coast, while the sea ice from north of Greenland and Canadian Arctic Archipelago to the pole remains relatively stable (Fig. 4c of Roach and Blanchard-Wrigglesworth 2022).

Owing to the distinct seasonal and spatial variations in Arctic sea ice as mentioned above, current research predominantly targets the sea ice decline in the Arctic marginal seas in summer and autumn (Bi et al., 2023; Kim et al., 2014; Liang and Zhou, 2023; Yu and Zhong, 2018). In contrast, the central Arctic, characterized by more stable sea ice conditions,

has garnered less attention and presents greater challenges for investigation. Unfortunately, the perennial sea ice in the central Arctic has begun to undergo extreme reductions in recent years (Comiso et al., 2008; Holland and Stroeve, 2011; Huang et al., 2021; Kwok and Rothrock, 2009; Li et al., 2018; Petty et al., 2018; Zhao et al., 2018), which lowers the recoverability of Arctic sea ice and enhances the sensitivity of Arctic sea ice to climate change (Li et al., 2022). Previous studies on sea ice in the central Arctic mainly focused on case analysis or long-term trends. Moreover, most prior research

concentrated on summer and autumn. This is because the sea ice in the central Arctic region is predominantly multi-year thick ice, and the absolute value of sea ice variation in winter and spring is relatively small. Consequently, it is difficult to systematically and comprehensively clarify the main spatiotemporal characteristics and mechanisms of the variations in the central Arctic sea ice. Overall, there is still a lack of common understanding regarding the changes in the central Arctic sea ice. This paper focuses on the ELSEs in the central Arctic. Departing from prior research, our analysis is grounded in an

expansive dataset spanning 34 years (1989–2022) of daily SIC measurements. Instead of focusing on specific seasons, we investigate the common characteristics and mechanisms of all ELSEs in the central Arctic. Thus, we focus on the relative variations of the ELSEs in the central Arctic, which have advantages in emphasizing the common spatial patterns of low SIC events of varying intensity and season.

The outline for the following sections is as follows. The second section describes the datasets and analytical methods

used in this study. The third section introduces the two dominant modes that contribute most to the variance in ELSEs. The fourth and fifth sections examine the thermodynamic and dynamic mechanisms and discuss the possible feedbacks. The sixth section discusses the teleconnection of the first mode. Finally, the last section synthesizes the key findings and provides a discussion of the results.

## 2 Data and methods

### 2.1 Sea ice data

We use daily 25 km × 25 km SIC (DiGirolamo et al., 2022) and sea ice motion (SIM) (Tschudi et al., 2019a) data from 1989 to 2022, provided by the National Snow and Ice Data Center (NSIDC), to explore the ELSEs. These datasets integrate data from multiple sensors and buoys and are combined with the NCEP/NCAR reanalysis data, to produce outputs that have been updated with errors well corrected. Additionally, the sea ice age data with spatiotemporal resolution of 7 days and 12.5 km ×

12.5 km, offered by NSIDC (Tschudi et al., 2019b), provides valuable insights into the distribution patterns of both multiyear and first-year ice in the Arctic.



The Pan-Arctic Ice Ocean Modeling and Assimilation System (PIOMAS) employs a coupled sea ice and ocean model to deliver refined reanalysis data of sea ice (Zhang and Rothrock, 2003), with an average spatial resolution of 22km (0.8°). This paper utilizes its daily data on sea ice thickness (SIT), SIC, and SIM for the diagnosis of the sea ice budget.

## 2.2 Atmospheric data

The atmospheric variables, with a spatial resolution of 0.25° × 0.25°, are primarily derived from the fifth version of European Centre for Medium-Range Weather Forecasts (ERA5) reanalysis dataset. The daily variables include temperature, wind speed, geopotential height, and specific humidity at various pressure levels (Hersbach et al., 2023a), as well as low cloud cover below 2 km and atmospheric surface boundary radiation flux, sensible heat flux, latent heat flux (Hersbach et al., 2023b).

The Japanese 55-year Reanalysis (JRA-55) data (Kobayashi et al., 2015) are used to calculate atmospheric heating rates: large-scale condensation heating, convective heating, vertical diffusion heating, solar radiation heating, and longwave radiation heating. This serves as the direct estimation of the diabatic heating rate to verify the accuracy of the rate calculated based on the residual term of temperature tendency equation in Sect. 2.4. The JRA-55 datasets have a temporal resolution of 6 hours and a spatial resolution of 1.25° by 1.25°.

## 2.3 General methods

The absolute variations are represented by the departures of each variable from its climatological mean. The relative variations are indicated by the standardization of these departures, derived by dividing the anomalies by their standard deviation. Here, we define ELSEs as those where the absolute variations in SIC are smaller than negative 1.5 standard deviations (SDs) of climatological mean. Empirical Orthogonal Function (EOF) analysis (Lorenz, 1956) is a useful method for discerning the spatiotemporal characteristics of variables. By applying EOF to the standardized SIC anomaly fields of the ELSEs over the central Arctic, the first two principal modes of the relative variations of ELSEs are extracted. Subsequently, the principal components of these two modes are regressed against the standardized SIC anomaly fields north of 60°N.

To identify the ELSEs for each mode, a threshold of 1.5 SDs is employed. Cases when the principal components exceed (or fall below) their mean by more than 1.5 SDs are considered as positive (negative) events for composite analysis. For convenience, only the composite differences between positive and negative events are shown here. Spatial correlation is calculated for each spatial mode. The significance of composite and correlation analyses is assessed using the two-tailed Student's *t*-test (Bretherton et al., 1999; Wilks, 2016).

## 2.4 Temperature tendency equation diagnosis

The variations of air temperature *T* can be diagnosed using the temperature tendency equation, as shown in Eq. 1:

$$\frac{\partial T}{\partial t} = -\overrightarrow{V_h} \cdot \nabla_h T - (\gamma - \gamma_d)\omega \frac{RT}{pg} + \frac{1}{c_p}\frac{dQ}{dt} \tag{1}$$



$\overrightarrow{V_h}$ and $\nabla_h$ represent the horizontal wind vector and the Hamiltonian operator; $\gamma$ and $\gamma_d$ represent the moist adiabatic and dry adiabatic lapse rates; $t, \omega, R, p, g, c_p, Q$ represent time, vertical velocity in the p-coordinate system, specific gas constant, air pressure, gravitational acceleration, specific heat at constant pressure, and heat content, respectively. The first term on the right side of the Eq. 1 represents the horizontal advection term, the second term denotes the combined effect of convective and adiabatic heating, and the third term represents the contribution of diabatic heating. The diabatic heating term is calculated as residual (Yanai et al., 1992; Yao and Sun, 2016), which transforms Eq. 1 into $\frac{1}{c_p}\frac{dQ}{dt} = \frac{\partial T}{\partial t} + \overrightarrow{V_h} \cdot \nabla_h T + (\gamma - \gamma_d)\omega\frac{RT}{pg}$. The diabatic heating rate can also be directly estimated by summing the large-scale condensation heating rate, convective heating rate, vertical diffusion heating rate, solar radiation heating rate, and longwave radiation heating rate based on the JRA-55 datasets.

**2.5 Sea ice budget**

Effective SIT $(H_{eff})$, the product of SIT and SIC, is used for sea ice budget analysis. The variation in $H_{eff}$ is caused by thermodynamic and dynamic forces, which can be expressed as Eq. 2,

$$\frac{\partial H_{eff}}{\partial t} = THE + DYN \tag{2}$$

where the $THE$ and $DYN$ represent the thermodynamic and dynamic terms, respectively. The dynamic term is composed of the horizontal advection term and the divergence term,

$$DYN = -\nabla(\boldsymbol{u}H_{eff}) = -\boldsymbol{u}\nabla H_{eff} - H_{eff}(\nabla\boldsymbol{u}) \tag{3}$$

where $\boldsymbol{u}$ represents the SIM vector. Then, the thermodynamic term is obtained as residual as shown in Eq. 4.

$$THE = \frac{\partial H_{eff}}{\partial t} - DYN \tag{4}$$

**2.6 Sea ice dynamic parameters**

Sea ice area flux, $F_{SIC}$, is given by Eq. 5,

$$\begin{cases} F_{SICx} = SIC \cdot udy \\ F_{SICy} = SIC \cdot vdx \end{cases} \tag{5}$$

$F_{SICx}$ and $F_{SICy}$ represent the zonal and meridional components of $F_{SIC}$; $u$ and $v$ denote the zonal and meridional SIM; $dx, dy$ are the incremental distances in the zonal and meridional directions, respectively. The divergence of $F_{SIC}$, denoted as $D_F$, is shown in Eq. 6,

$$D_F = \frac{\partial F_{SICx}}{\partial x} + \frac{\partial F_{SICy}}{\partial y} \tag{6}$$

the positive directions of $x$ and $y$ are eastward and northward, respectively.

According to Helmholtz decomposition, the velocity field can be decomposed into divergent and rotational components,



$$\mathbf{V} = \mathbf{V}_\chi + \mathbf{V}_\psi \tag{7}$$

The potential function $\chi$ and stream function $\psi$ satisfy

$$\begin{cases} D = \dfrac{\partial u}{\partial x} + \dfrac{\partial v}{\partial y} = \dfrac{\partial(u_\chi + u_\psi)}{\partial x} + \dfrac{\partial(v_\chi + v_\psi)}{\partial y} = -\nabla^2 \chi \\[2mm] \zeta = \dfrac{\partial v}{\partial x} - \dfrac{\partial u}{\partial y} = \dfrac{\partial(v_\chi + v_\psi)}{\partial x} - \dfrac{\partial(u_\chi + u_\psi)}{\partial y} = \nabla^2 \psi \end{cases} \tag{8}$$

$D, \zeta$ represent divergence and relative vorticity; $u, v$ denote zonal and meridional velocities of the flow field; the Hamiltonian operator is denoted by $\nabla$. Here we use the spherical H-C method (Dawson, 2016) to decompose the SIM into divergent and rotational fields.

This paper also accounts for sea ice deformation, including stretching and shear deformations,

$$\begin{cases} stretching\ deformation = \dfrac{\partial u}{\partial x} - \dfrac{\partial v}{\partial y} \\[2mm] shear\ deformation = \dfrac{\partial u}{\partial y} + \dfrac{\partial v}{\partial x} \end{cases} \tag{9}$$

$$total\ deformation = \sqrt{\left(\dfrac{\partial u}{\partial x} - \dfrac{\partial v}{\partial y}\right)^2 + \left(\dfrac{\partial u}{\partial y} + \dfrac{\partial v}{\partial x}\right)^2} \tag{10}$$

**2.7 T-N wave flux and Rossby wave source**

The T-N wave flux effectively captures the propagation pathways of atmospheric Rossby waves (Takaya and Nakamura, 2001), serving as a useful tool for analyzing teleconnections between mid-latitude and polar regions. The calculation formula is as follows:

$$W_x = \frac{p\cos\phi}{2|\mathbf{U}|}\left\{\frac{U}{a^2\cos^2\phi}\left[\left(\frac{\partial\psi'}{\partial\lambda}\right)^2 - \psi'\frac{\partial^2\psi'}{\partial\lambda^2}\right] + \frac{V}{a^2\cos\phi}\left[\frac{\partial\psi'}{\partial\lambda}\frac{\partial\psi'}{\partial\phi} - \psi'\frac{\partial^2\psi'}{\partial\lambda\partial\phi}\right]\right\} \tag{11}$$

$$W_y = \frac{p\cos\phi}{2|\mathbf{U}|}\left\{\frac{U}{a^2\cos\phi}\left[\frac{\partial\psi'}{\partial\lambda}\frac{\partial\psi'}{\partial\phi} - \psi'\frac{\partial^2\psi'}{\partial\lambda\partial\phi}\right] + \frac{V}{a^2}\left[\left(\frac{\partial\psi'}{\partial\phi}\right)^2 - \psi'\frac{\partial^2\psi'}{\partial\phi^2}\right]\right\} \tag{12}$$

$p = pressure/1000hPa$; $\mathbf{U} = (U, V, 0)^T$ represents the background flow field; $\phi, \lambda, a, f$ denote latitude, longitude, Earth radius, and the Coriolis parameter, respectively; $\psi = \phi/f$ is the geostrophic stream function.

By diagnosing the Rossby wave source (RWS) (Sardeshmukh and Hoskins, 1988), the generation sites of Rossby waves can be investigated,

$$RWS = -\nabla \cdot (\mathbf{V}_\chi \zeta_a) = -\zeta_a \nabla \cdot \mathbf{V}_\chi - \mathbf{V}_\chi \cdot \nabla \zeta_a \tag{13}$$

$\zeta_a$ represents the absolute vorticity, and $\mathbf{V}_\chi$ denotes the divergent wind. The first term on the right-hand side corresponds to the vortex stretching, while the second term signifies the advection of absolute vorticity by divergent wind.



## 3 Dominant modes

### 3.1 Indicative significance of sea ice in the central Arctic

Due to different research foci and objectives, the definition of the central Arctic varies. For convenience, previous studies
often defined the central Arctic from a geographical perspective, such as designating areas north of 74°N (Han et al., 2023),
80°N (Cai et al., 2021a; Laliberté et al., 2016; Zhao et al., 2018), or 84°N (Li et al., 2018) as the central Arctic. Politically,
Van Pelt et al. (2017) defined the waters north of the exclusive economic zone boundaries of the five Arctic coastal states as
the Central Arctic International Waters. To reveal the distributional features of Arctic cyclones, Kong et al. (2024) defined
the cyclone-active region within the Arctic Ocean, the Alpha Ridge area (78°N–90°N, 150°E–180°–90°W), as the central
Arctic. Ji and Zhao (2015) defined the general range of high SIC areas as the central Arctic to study the relationship between
SIC and clouds. Huang et al. (2021) delineated areas with a climatological SIC greater than 90% as the central Arctic. This
study applies the methodology of Huang et al. (2021) to define the central Arctic as the area where the climatological SIC
from 1989 to 2022 exceeds 90% (Fig. 1a). Figure 1b indicates that the SIC within the defined central Arctic exhibits minimal
seasonal variation, suggesting that this area predominantly consists of multi-year and thick ice (Fig. S1). It should be noted
that areas with SIC greater than 90% vary over time (blue lines in Fig. S2), and they are significantly positively correlated
with the mean SIC in the central Arctic as defined in this study and with the Arctic sea ice extent. Furthermore, the central
Arctic defined here well encompasses the high-probability areas where SIC exceeds 90% across all seasons (Fig. S3), which
validates the rationality of the definition of the central Arctic in this study.

Despite the relative stability of SIC within the central Arctic, both the intensity and frequency of the ELSEs have
significantly increased over the past two decades (Figs. 1c, e and S4b). The correlation coefficient between SIC anomaly in
the central Arctic and SIC anomaly in the entire Arctic is 0.61 (Fig. 1c), significant at the 0.01 significance level. Thus, the
fluctuations of SIC in the central Arctic are indicative of changes of SIC in the entire Arctic. Moreover, high correlations
occur when SIC anomaly in the central Arctic precedes SIC anomaly in the marginal Arctic within a two-month window,
whereas the correlation coefficients decrease sharply when SIC anomaly in the central Arctic lags behind SIC anomaly in the
marginal Arctic (Fig. S5). This suggests that a decrease of SIC in the central Arctic has a facilitating effect on subsequent
marginal Arctic SIC changes over the next two months.

### 3.2 ELSEs in the central Arctic

The ELSEs correspond to SIC values below the light grey shaded region in Fig. 1d. The lower boundary of this region
closely follows the 90th percentile of SIC in the central Arctic. Thus, the ELSEs denote an occurrence probability of roughly
10%, totaling 1022 days across 34 years. A notable upsurge of ELSEs has been detected after 2002, with no significant
seasonal disparities in the frequency of these events (Figs. 1e and S4). Nevertheless, the magnitude of the ELSEs during the
summer season is considerably more pronounced than in other seasons (Fig. 1e).





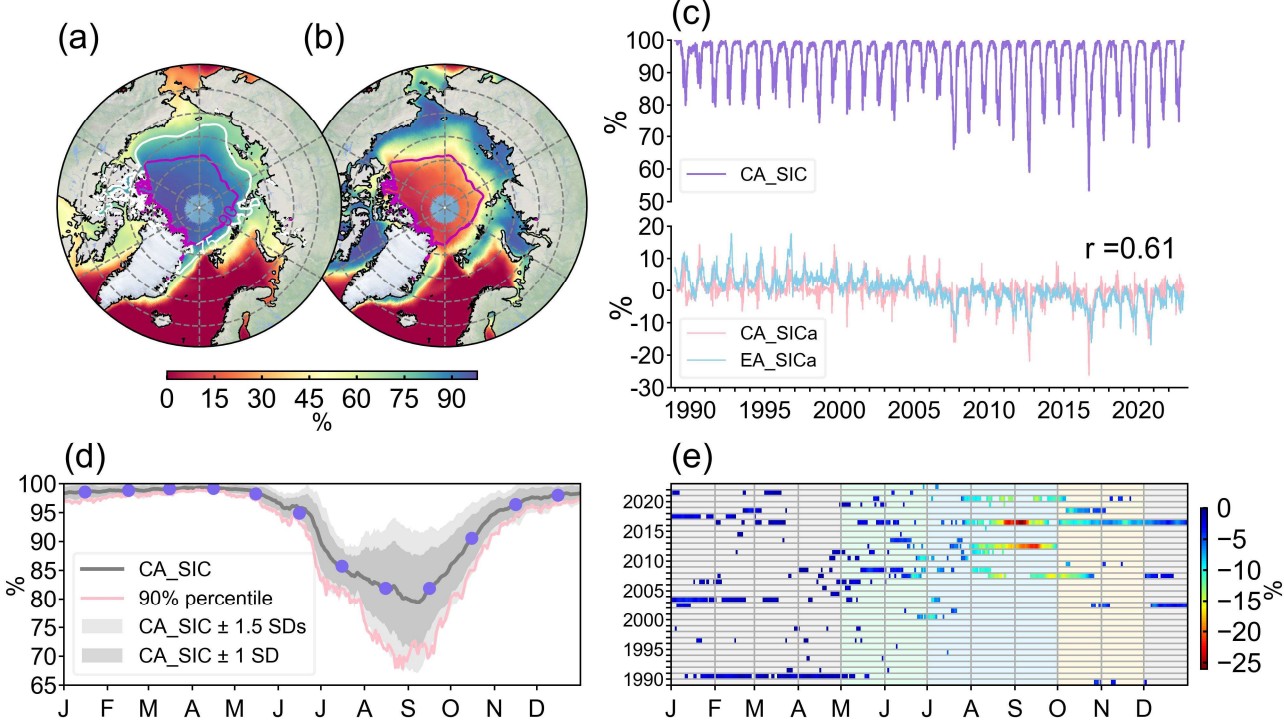

**Figure 1. (a)** Climatological SIC from 1989 to 2022 and **(b)** its seasonal difference (i.e., March minus September). Magenta and white curves represent the 90% and 75% SIC contours, respectively. **(c)** Time series of central Arctic SIC (CA_SIC), central Arctic SIC anomaly (CA_SICa), and entire Arctic SIC anomaly (EA_SICa). Letter r denotes the correlation coefficient between CA_SICa and EA_SICa. **(d)** Climatological seasonal variation of CA_SIC (gray curve and purple circles) and the 90th percentile of CA_SIC (pink solid line). The dark (light) gray shading represents the ranges within 1 (1.5) SD of CA_SIC. **(e)** Seasonal and interannual distribution of ELSEs in the central Arctic. Gray, green, blue, and golden fillings represent the winter, spring, summer, and autumn seasons defined by SIC seasonal variation according to d). Shadings represent the magnitude of SIC anomaly.

### 3.3 East–West Seesaw Mode and Pacific–Atlantic Seesaw Mode

In terms of the SIC anomaly of the ELSEs, the SIC anomaly in summer is often much greater than that of other seasons (Fig. 1e). Consequently, applying EOF method to the spatiotemporal field of SIC anomalies yields dominant modes that predominantly reflect the signals of summer SIC anomalies, which spatially manifest as significant SIC anomaly signals along the Arctic marginal seas and the edge of the central Arctic (not shown). This finding is spatially consistent with the result obtained by Yu and Zhong (2018), which is based on the Self-Organizing Maps method (their Fig. 1).



In contrast to original SIC anomaly without standardization (Fig. 1e), the standardized results represent the relative SIC variation in the central Arctic and exhibit significant signals across all seasons (Fig. 2a). EOF analysis of the standardized

SIC anomaly in the central Arctic reveals the first two dominant modes: the East–West Seesaw Mode (EWSM) and the Pacific–Atlantic Seesaw Mode (PASM), with variance contributions of 20.18% and 12.85%, respectively (Fig. 2b and c). The EWSM is characterized by an out-of-phase relationship of SIC anomaly in the central Arctic between the Eastern and Western Hemispheres, while the PASM manifests an out-of-phase relationship of SIC anomaly in the central Arctic between the Pacific and Atlantic sectors. Both the EWSM and the PASM pass the North test (North et al., 1982), indicating that they

are independent modes of the standardized SIC anomalies. Screening the time coefficients of each mode using a threshold of 1.5 SDs, the strong events of EWSM are primarily concentrated in 2002, 2003 and 2007; the PASM is mainly focused on the years 1990, 2007, 2008, and 2016.

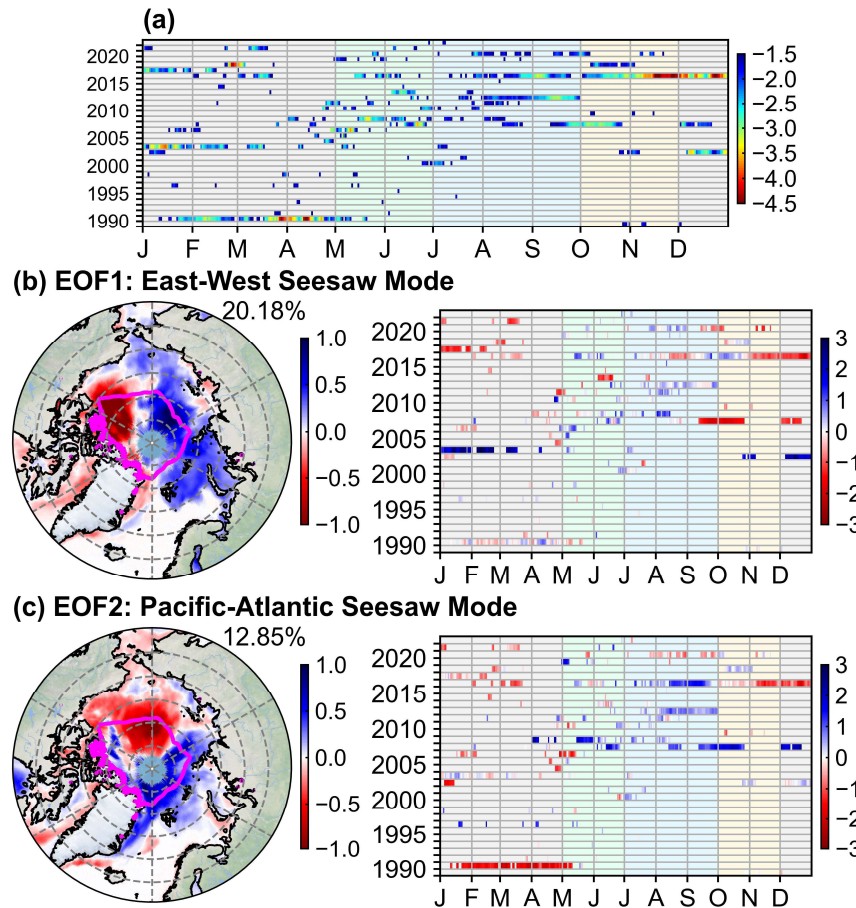

**Figure 2. (a)** Same as Fig. 1e but for standardized results. Principal components (right) and regression fields of principal

components against entire Arctic SIC anomalies (left) for **(b)** the EWSM and **(c)** the PASM. Principal components have been standardized. The variance contribution is displayed at the top right corner of the regression fields. The magenta curve represents the central Arctic.



## 4 Thermodynamic effect

### 4.1 Significance of diabatic heating for EWSM and PASM

According to the sea ice budget, the thermodynamic contribution accounts for approximately 67.8% and 72.2% (Fig. 3), indicating that the formation of both EWSM and PASM is thermodynamically dominated. The spatial correlation coefficient between the EWSM (PASM) and the surface air temperature is -0.77 (-0.91) (Fig. 4). Figure 5 presents the diabatic heating anomalies calculated based on the temperature tendency equation. In the central Arctic, the out-of-phase anomalies of diabatic heating between the Eastern and Western Hemispheres are robust (Fig. 5a). The most significant signals are located

near 120°W and 120°E, which is consistent with Figs. 2b and 4a. Figure. 5b shows an increase of diabatic heating anomalies in the Pacific sector and a decrease in the Atlantic sector, which corresponds well with Figs. 2c and 4b. We also estimate the diabatic heating anomalies using the JRA-55 datasets (Fig. S6), which are generally consistent with those in Fig. 5. The above analysis suggests that local surface temperature anomalies caused by local diabatic heating anomalies are the important factors in the formation of EWSM and PASM. Furthermore, Fig. S7 indicates that latent heat flux anomalies are

prominent in the formation of the EWSM and PASM.

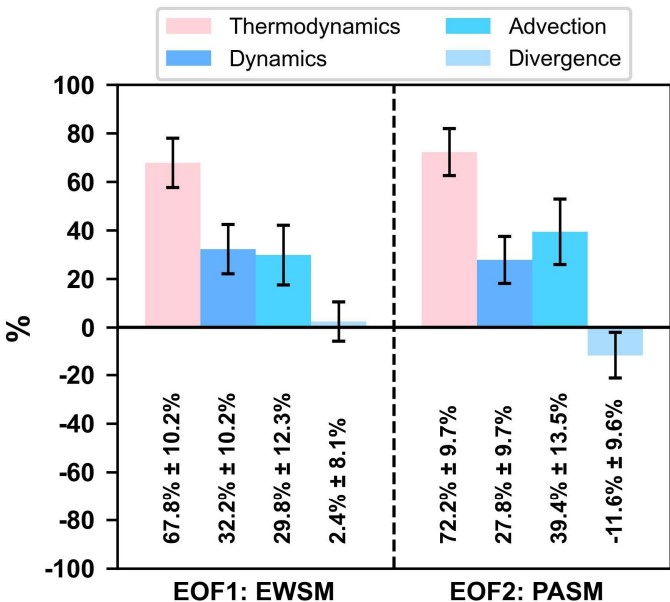

**Figure 3.** Effective SIT budget for the EWSM (left) and the PASM (right). Error bar represents the range of 1 SD from the mean value.

Local temperature fluctuations are influenced not only by diabatic heating but also by convective heating, adiabatic heating, and temperature advection. In the process of calculating diabatic heating, we naturally calculate the convective-



adiabatic heating term and the temperature advection term (not shown). At the near-surface level (1000 hPa), the convective-adiabatic heating term is approximately an order of magnitude smaller than the diabatic heating term, and thus can be neglected. Numerous studies have indicated that warm and moist air flows from mid-latitudes significantly affect Arctic

warming and sea ice reduction (Liang and Zhou, 2023; Liang et al., 2023; Yang and Magnusdottir, 2018; Zhang et al., 2023). However, the temperature advection term contributes insignificantly to the formation of the EWSM and the PASM. For the EWSM, the meridional extension of geopotential height anomalies from mid-latitudes to the polar region is weak (Fig. 6a and c), resulting in minor influence from warm and moist air flows from mid-latitudes. In terms of the PASM, an anomalous high pressure near Alaska and an anomalous low pressure to its west lead to strong anomalous poleward winds (Fig. 6b and

d). According to previous studies, the deepening of the Aleutian Low facilitates the transport of warm and moist air into the Arctic, leading to sea ice loss primarily in the Pacific sector and even extending to the central Arctic Ocean (England et al., 2020; Screen and Deser, 2019; Svendsen et al., 2018). However, a strong ridge invades the Arctic from the Greenland Sea, causing the polar vortex to split into a double-vortex structure (Fig. S8b). The central Arctic is covered by a barotropic anomalous high-pressure system (Figs. S9d–f and S10a–d). This configuration prevents the transport of warm and moist air

from the Pacific to the central Arctic, thus hindering the influence of temperature advection on the central Arctic.

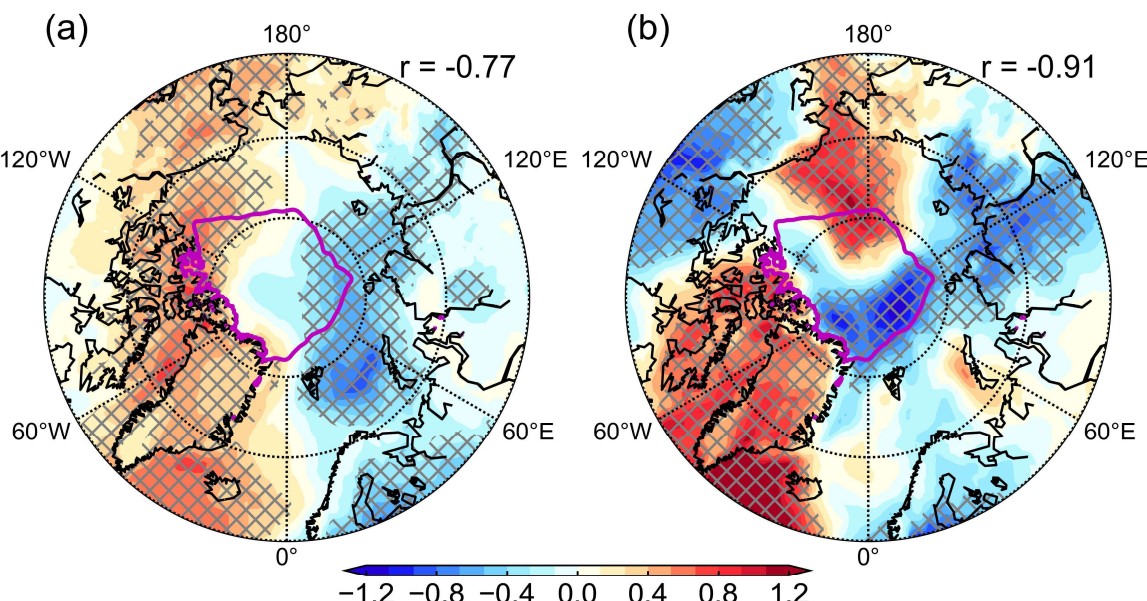

**Figure 4.** Composite differences in standardized surface air temperature anomaly between the positive and negative phases of **(a)** the EWSM and **(b)** the PASM. Magenta curve represents the central Arctic. The spatial correlation coefficients with the corresponding spatial patterns (Fig. 2b, c) in the central Arctic are shown in the upper right corner. The grey crossings

indicate areas with significant difference at the 0.05 significance level based on Student's *t*-test.



## 4.2 Water vapor and cloud feedbacks

Typical events of the EWSM and PASM are predominantly observed during the winter and spring seasons (Fig. 2b and c). At this time, weak solar radiation in the central Arctic results in weak surface albedo feedback. The reduction of sea ice favors an increase in water vapor, hence the patterns of near-surface water vapor variations are almost identical to those of

the EWSM and the PASM (Fig. 7a and e). Changes in water vapor in the lower troposphere are more critical for water vapor feedback, which are closely linked to variations in low cloud cover. The impacts of water vapor and clouds on surface temperature depends on the competition between shortwave and longwave radiation (Hu et al., 2018; Kapsch et al., 2016). The spatial distribution of relative variations in lower tropospheric water vapor (Fig. 7b and f) and low cloud cover (Fig. 7c and g) results in the distribution of net radiative flux changes (Fig. 7d and h) that corresponds to the two dominant modes of

the ELSEs in the central Arctic (Fig. 2b and c). Hence, the water vapor and cloud feedbacks, induced by variations in water vapor and low clouds, facilitate the formation and development of the EWSM and PASM.

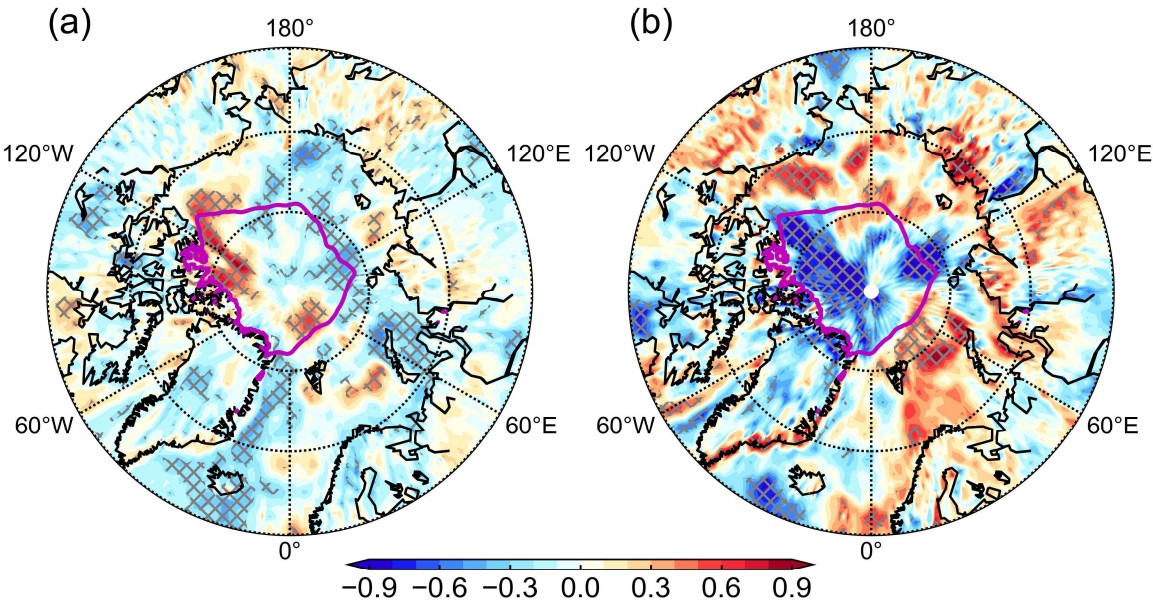

**Figure 5.** Same as Fig. 4 but for the diabatic heating anomalies.

## 5 Dynamic effect

In contrast to thermodynamic processes, the mechanical movement of sea ice can spatially redistribute it without directly causing melt or growth. Atmospheric circulation plays a fundamental role in causing Arctic sea ice anomalies by altering SIM, and the transportation of sea ice towards subpolar regions can lead to a decrease in Arctic sea ice (Cai et al., 2021b;



Jeong et al., 2022; Nghiem et al., 2007; Sumata et al., 2022). Furthermore, dynamic processes often couple with thermodynamic effects. For example, Zhao et al. (2018) found that the divergence associated with positive wind stress curl

increases the open water area in the central Arctic, triggering a surface albedo feedback that intensifies sea ice reduction there. Both the fragmented ice caused by ocean wave action and the thin ice formed by thermodynamic processes are more susceptible to wind stress forcing, resulting in more open water areas that absorb solar radiation (Voosen, 2020). The stretching and shearing motions of sea ice contribute to the reduction of ice thickness and the formation of ice leads (Bi et al., 2023), and the accumulation of ice lead numbers and areas during winter and spring can significantly exacerbate the

subsequent summer melt of sea ice (Schröder et al., 2014). The coupling of dynamic and thermodynamic processes means that dynamic effects in the EWSM and PASM cannot be ignored.

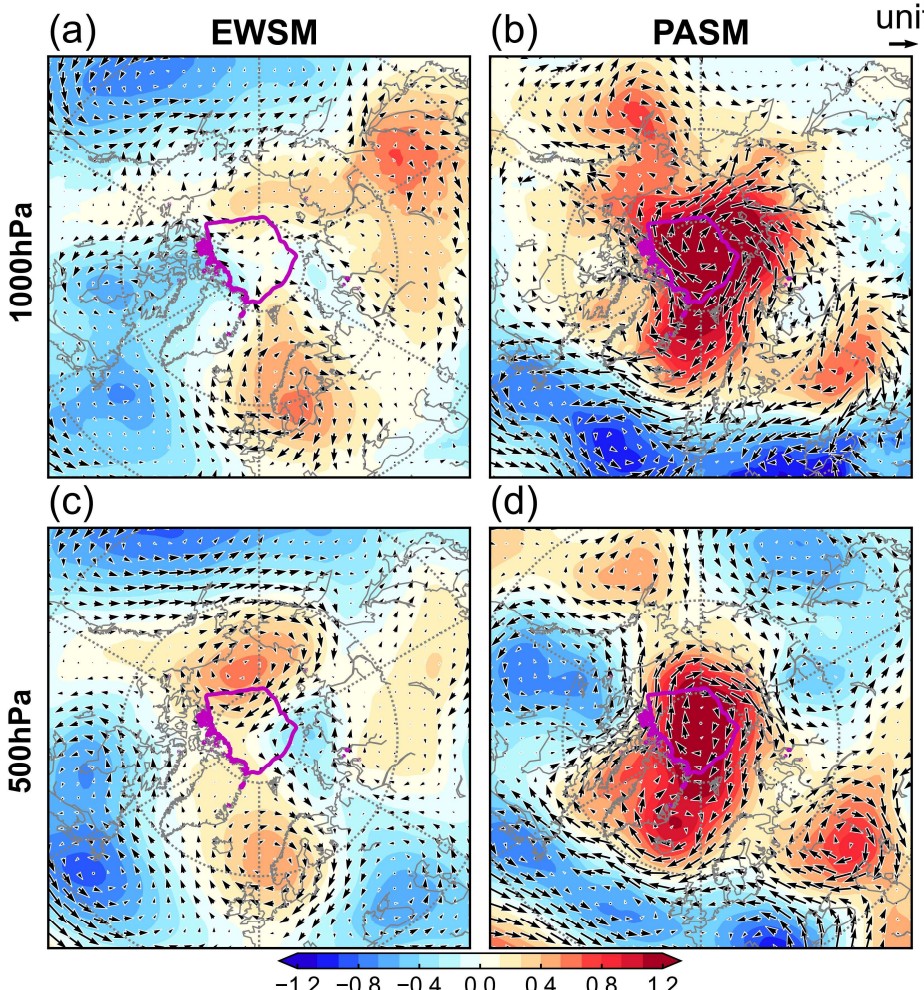

**Figure 6.** Composite differences between the positive and negative phases of the standardized geopotential height (shadings) and wind (arrows) anomalies for **(a, c)** the EWSM and **(b, d)** the PASM. The upper and lower rows show results for the 1000

hPa and 500 hPa pressure levels, respectively. The magenta line denotes the central Arctic.



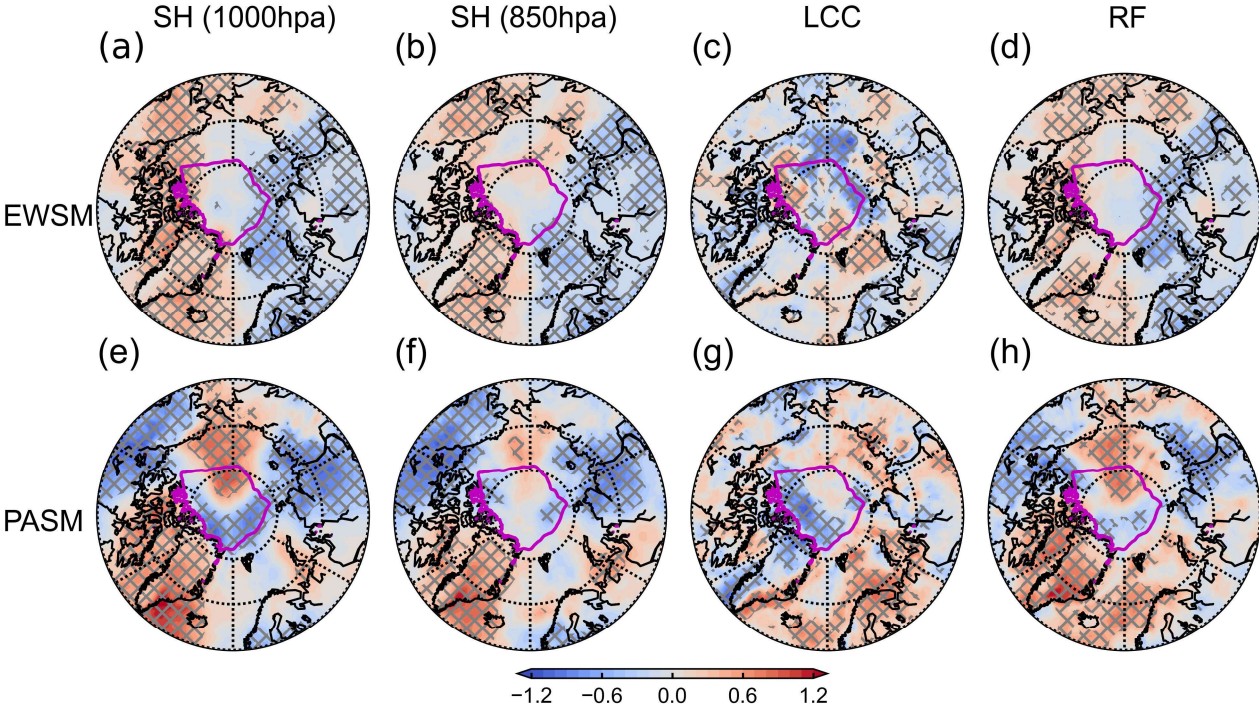

**Figure 7.** Composite differences between positive and negative phases of the standardized anomalies of **(a, e)** 1000 hPa specific humidity, **(b, f)** 850 hPa specific humidity, **(c, g)** low cloud cover, and **(d, h)** net longwave and shortwave radiation flux at the atmospheric surface boundary for **(a–d)** the EWSM and **(e–h)** the PASM. Grey crossings denote areas with significant difference at the 0.1 significance level based on Student's t-test. Magenta lines represent the central Arctic.

Over 70% of SIM is attributed to geostrophic winds (Thorndike and Colony, 1982). Hence, the standardized SIM anomalies are essentially consistent with the standardized wind anomalies (Figs. 6 and 8). By decomposing the standardized SIM fields into divergence (Fig. 9a and b) and vorticity (Fig. 9c and d) components, it can be observed that the sea ice advection primarily drives the anomalies in sea ice motion, serving as the main source of dynamic contribution in the formation processes of both the EWSM and the PASM (Fig. 3). There is an out-of-phase relative variation of $D_F$ between the Eastern and Western Hemispheres composited for EWSM mode (Fig. 8a), with convergence and divergence zones corresponding to the abnormal increase and decrease of sea ice. This indicates that the convergence and divergence of sea ice positively contribute to the formation of the EWSM (Fig. 3). The PASM is primarily characterized by sea ice divergence, with some areas of sea ice convergence in the Pacific sector (Fig. 8b). This spatial pattern is opposite to that of the PASM, indicating a compensatory role for the formation of PASM (Fig. 3).

Interestingly, the unique dynamic processes of the two modes, particularly the anomalous sea ice advection, pose a further potential threat to the reduction of Arctic sea ice. For the EWSM, the anticyclonic anomaly east of Iceland and the cyclonic anomaly northeast of Canada result in significant ice transport from the multi-year ice area (Fig. S1) towards the



Beaufort Sea (Figs. 6a, 8a, and 9c). Due to the high-pressure anomaly located in the Beaufort Sea and the low-pressure

anomaly located in the Kara Sea, the thick ice enters the Beaufort Sea and is transported further into the Eastern Hemisphere (Figs. 6a, 8a, and 9c). This advective effect is crucial for the spatial distribution of Arctic sea ice, as Labe et al. (2018) found an antiphase dipole pattern of SIT change between the East Siberian Sea and the Fram Strait based on PIOMAS and the Community Earth System Model Large Ensemble Project datasets. The outward transport of multi-year thick ice suggests the potential of EWSM to diminish the stability of Arctic sea ice, which may explain the increased frequency and intensity of

the ELSEs after 2002 (the largest year for the EWSM; Fig. 1c and e). Furthermore, the Arctic Amplification underwent an interdecadal transition in 2002 (Wang et al., 2017), which might be a result of the unique dynamical processes of the EWSM coupled with thermodynamic effects.

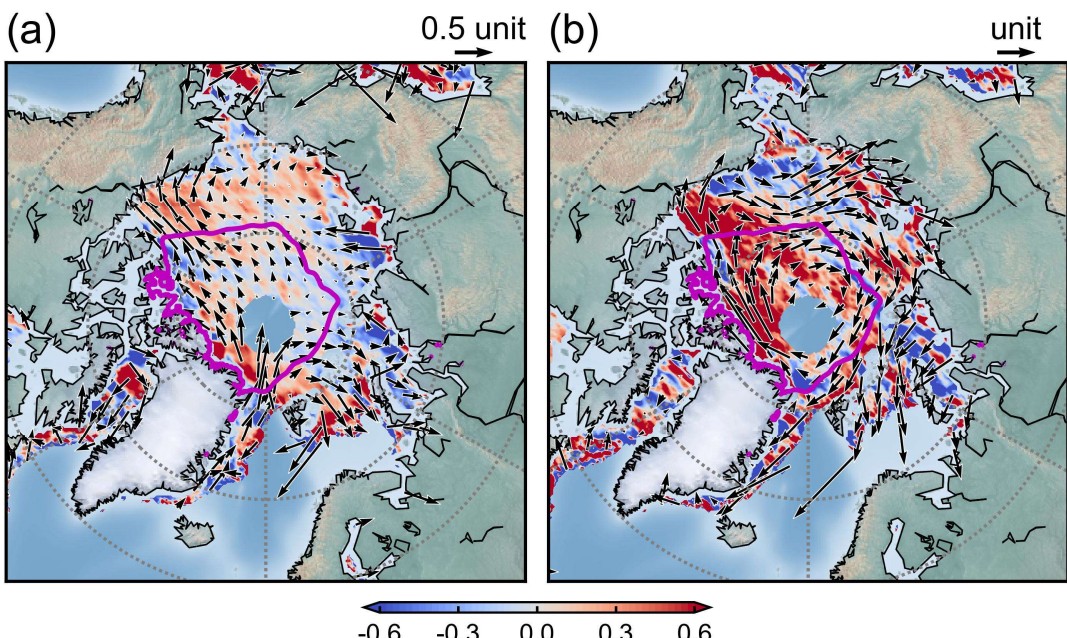

**Figure 8.** Composite differences between the positive and negative phases of the standardized sea ice area flux anomalies for

**(a)** the EWSM and **(b)** the PASM. Arrows indicate the sea ice area flux anomalies and shadings represent their divergences. The magenta line denotes the central Arctic. Note that the scales of the arrows differ.

Regarding the PASM, the anomalous sea ice advection induced by large-scale ridge dominates the variability in SIM. The anomalous advection generally follows a clockwise direction and passes through the Fram Strait and Barents Sea (Fig. 9d), resulting in sea ice increase in the Atlantic sector. It is noteworthy that the anomalous northward strong winds, induced

by the abnormal high pressure south of Alaska, and the clockwise anomalous circulation, are nearly perpendicular in the wind direction within the Pacific sector (Fig. 6b and d). This leads to deformation and even fragmentation of sea ice in the Pacific sector (Fig. S11). Consequently, the formation of wind ridges and ice leads is favored, with thinner ice being more





likely. The spatial pattern of Fig. S11 is consistent with the PASM (Fig. 2c), implying the sea ice deformation process may contribute to the maintenance and development of the PASM through dynamic and thermodynamic coupling feedback (Bi et al., 2023; Schröder et al., 2014; Voosen, 2020).

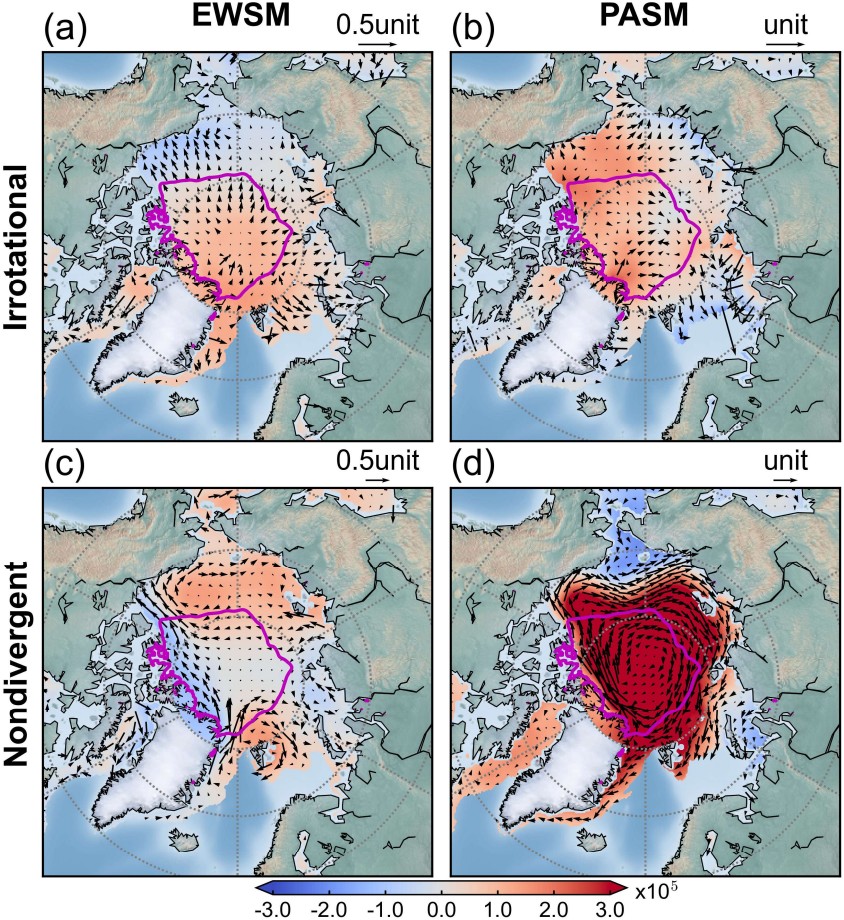

**Figure 9.** Composite differences between the positive and negative phases of the standardized divergent (arrows in **a, b**) and rotational (arrows in **c, d**) components of SIM anomalies for **(a, c)** the EWSM and **(b, d)** the PASM. Shadings in **(a, b)** represent the potential function, and shadings in **(c, d)** represent the stream function. Note that the scales of the arrows differ.

## 6 North Pacific–Arctic teleconnection

This section will explore the genesis of the atmospheric circulation anomalies that drive the advective transport associated with the EWSM. The EWSM exhibits prominent mid-to-high-latitude planetary wave trains (Fig. S9a–c). Previous studies have demonstrated that the mid-latitude and polar regions can interact through teleconnections (Baxter et al., 2019; Deng and Dai, 2024; Grunseich and Wang, 2016; Zhou et al., 2024). Considering that the most pronounced EWSM signal during the winter of 2002 (Fig. 2b) and the propagation speed of Rossby waves, we focus primarily on the autumn and winter of 2002.




Figure 10 illustrates that the upper-tropospheric geopotential height anomaly signals of the mid-latitudes and Arctic are linked by a barotropic wave train (Figs. 6a, c and S9a–c). In the polar regions, a positive dipole appears over the Pacific and Atlantic sectors. In the middle of the dipole, a negative centre forms over eastern North America and extends to northern Greenland and the Canadian Arctic Archipelago (Fig. 6c). These circulation features are key to influencing the sea ice

advection of the EWSM (Figs. 8a and 9c). Moore et al. (2022) observed a partial recovery of sea ice in the Beaufort Sea in 2021, which was not due to increased ice growth but rather the advection of thick sea ice from the north of Greenland and the Canadian Arctic Archipelago region. This illusion of local sea ice recovery through advection across thick ice areas represents an overdraw on Arctic sea ice, and the reduction of multi-year thick ice can enhance the Arctic air-ice coupling (Li et al., 2022; Williams et al., 2016).

In 2002, the aforementioned geopotential height anomalies were already prominent in the season preceding the EWSM (Fig. 10a), highlighting the close linkage between mid-latitude and Arctic regions via teleconnections. Further diagnostics indicate that the North Pacific is the source region for exciting the aforementioned teleconnections. In autumn, the negative RWS is primarily located near the Gulf of Alaska (Fig. 11a), and by winter, it moves southward to around 40°N and further intensifies (Fig. 11d). The RWS generation is predominantly attributed to vortex stretching (Fig. 11b, e), suggesting that the

upper tropospheric divergence anomalies over the North Pacific induce the RWS (Miao and Jiang, 2021).

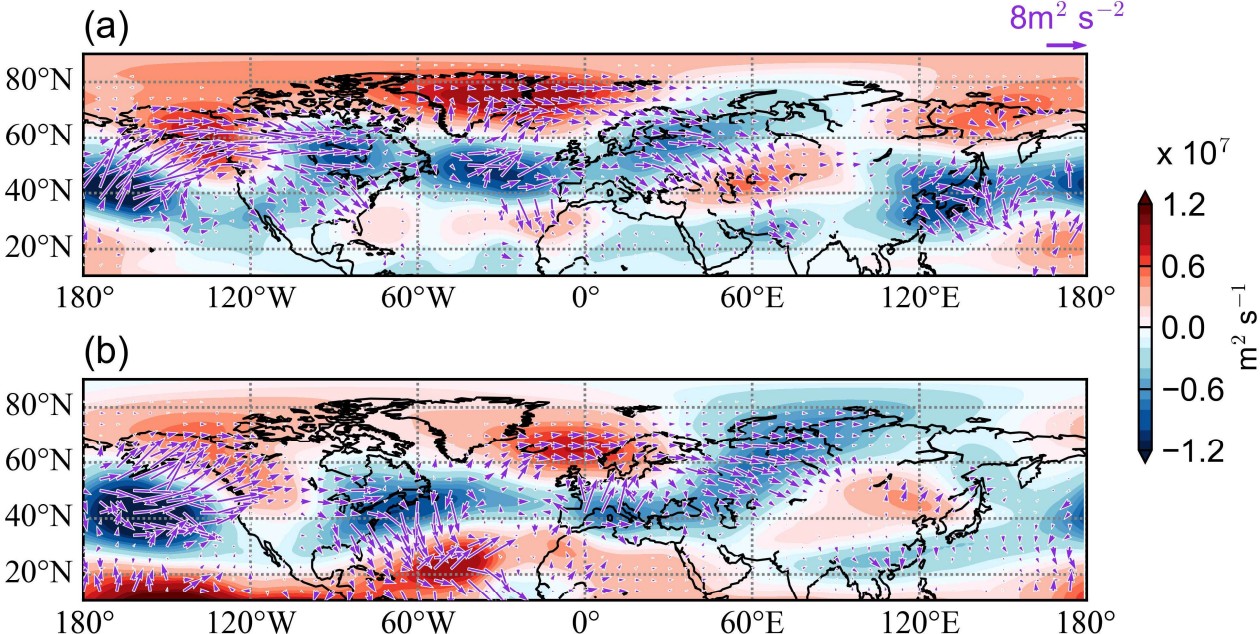

**Figure 10. (a)** Autumn (September–November) and **(b)** winter (following December–January) T-N wave activity flux (arrows) at 300 hPa in 2002. Shading represents the stream function.



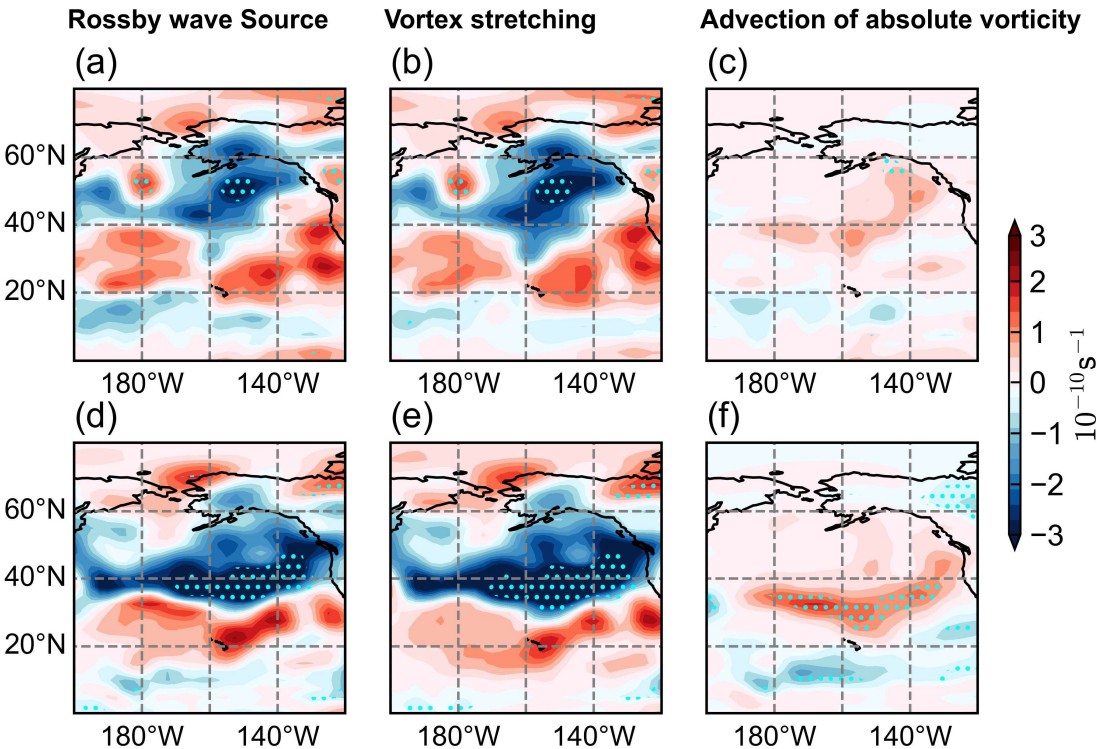

**Figure 11. (a)** RWS at 300 hPa, **(b)** the vortex stretching component, and **(c)** the advection of absolute vorticity component in autumn of 2002. **(d)–(f)** are same as **(a)–(c)**, but for the winter of 2002. Dots denote areas with significant difference at the 0.05 significance level based on Student's *t*-test.

## 7 Conclusions and discussions

This study defines the area where climatological SIC exceeds 90% as the central Arctic and designates the ELSEs where the average SIC in the central Arctic deviates more than 1.5 SDs below the climatology. Employing the EOF method, the first two dominant modes of the relative variations of ELSEs were identified. Figure 12 presents a schematic diagram of the formation mechanisms for the two modes. EOF1 exhibits out-of-phase SIC anomalies between the Eastern and Western Hemispheres in the central Arctic, termed the EWSM; EOF2 demonstrates out-of-phase SIC anomalies between the Atlantic and Pacific sectors in the central Arctic, termed the PASM. Thermodynamic forcing dominates the formation of the EWSM and PASM, which contributes approximately 68% and 72%, respectively. Moreover, variations in thermodynamic factors are primarily driven by local diabatic heating anomalies. Under thermodynamic dominance, both the EWSM and the PASM trigger water vapor and cloud feedbacks to sustain and enhance their development. The dynamic contribution accounts for 32% and 28% in the EWSM and PASM, respectively. For the EWSM, the thick ice from the north of Greenland and the Canadian



Arctic Archipelago region is strongly advected towards the Beaufort Sea. The thick ice enters the Beaufort Sea and is

transported further into the Eastern Hemisphere due to the high-pressure anomaly located in the Beaufort Sea and the low-pressure anomaly located in the Kara Sea. This process that couples with thermodynamic processes will make Arctic sea ice more sensitive. The related atmospheric circulation anomalies are caused by barotropic teleconnections generated by the upper tropospheric divergence anomaly forcing in the North Pacific. Additionally, the convergence and divergence of SIM exert a minor positive contribution to the EWSM. For the PASM, the influence of the Atlantic-invasive ridge and the

Alaskan high-pressure anomaly results in a roughly heart-shaped SIM anomaly field. Intense sea ice advection anomalies lead to an increase in sea ice anomalies in the Atlantic sector and a decrease in the Pacific sector. Meanwhile, the convergence and divergence of sea ice contribute negatively. An intriguing phenomenon is that the coupling of dynamically-induced sea ice deformation with thermodynamic effects may facilitate the formation of the PASM. The dynamical processes of the EWSM and PASM both reflect multi-factor feedback mechanisms involving dynamic and thermodynamic

coupling. As sea ice melts and its thickness decreases, it becomes more susceptible to forced deformation or fracturing. The roughened surface of fragmented ice is more affected by wind stress, which leads to an increase in open water areas. The expansion of open water areas not only allows the ocean to absorb more solar radiation but also makes it more directly subject to wind influence, leading to enhanced ocean currents and waves. Therefore, the dynamical effects are intensified. The end result is further reduction of sea ice, forming a positive feedback loop (Voosen, 2020).

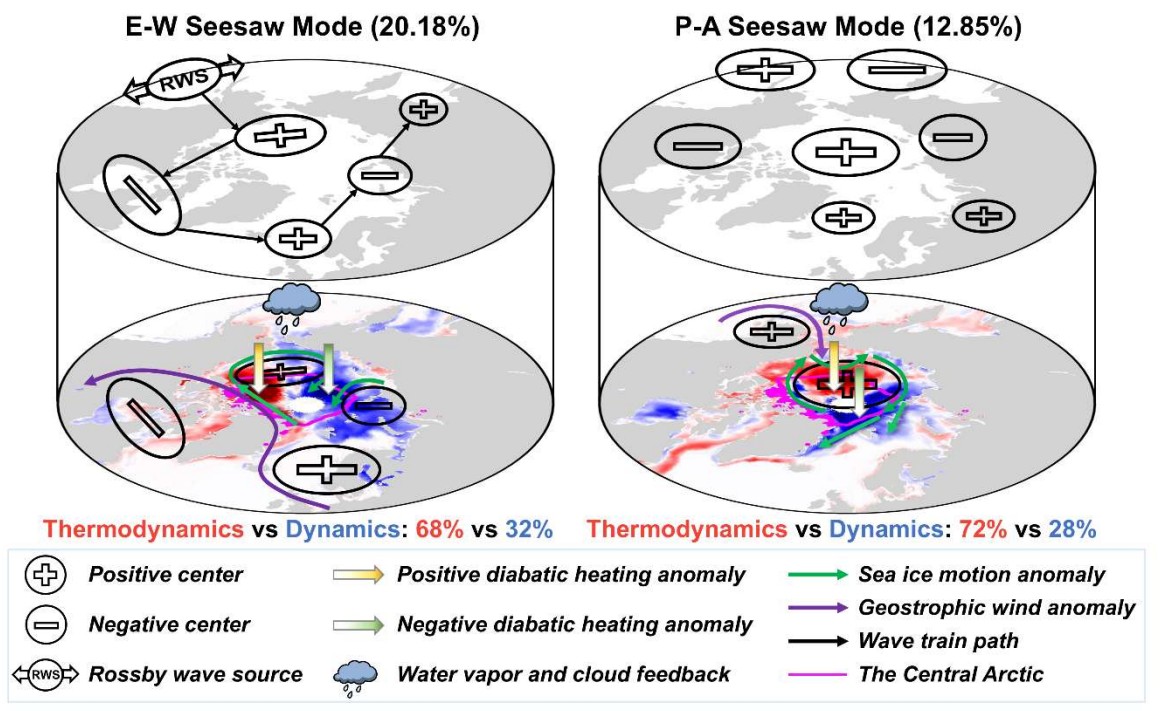


**Figure 12.** Schematic diagram of the formation mechanisms for the EWSM (left) and the PASM (right).



It is crucial for sea ice prediction to recognize the role of large-scale atmospheric forcing and circulation anomalies in Arctic climate change (Topál and Ding, 2023). Cai et al. (2021b) noted that Arctic sea ice has certain connections with the Arctic Oscillation, Atlantic Multidecadal Oscillation, North Atlantic Oscillation, Pacific Decadal Oscillation, and Arctic
Dipole. This study just highlights the significance of the North Pacific–Arctic teleconnection for the EWSM. The relationship between this teleconnection, as well as the EWSM and PASM, with large-scale internal variability remains to be explored in detail in the future. Such investigation is essential for improving models and Arctic sea ice prediction.

Liang et al. (2022b) have indicated that the state of Arctic sea ice is primarily governed by local mean atmospheric conditions. For instance, the intensification of a high-pressure ridge over the Arctic Ocean promotes subsidence in the lower
troposphere and consequently induces adiabatic heating (Ding et al., 2019; Papritz, 2020; Sedlar and Tjernström, 2017; Serreze et al., 2019). Similarly, the atmospheric circulation anomaly associated with the PASM manifests as a strong barotropic ridge that invades the Arctic from the Atlantic (Figs. S8b and S10a–d). This leads to a coherent warming anomaly from 850 hPa to 200 hPa over the central Arctic due to adiabatic sinking (Fig. S10f–h). However, surface temperature in the Atlantic sector is anomalously low at 1000 hPa (Figs. 3b and S10e), resulting in increased SIC. In Sect. 4.1, we concluded
that the PASM is primarily driven by anomalous diabatic heating. Moreover, Fig. 5b illustrates that the cooling effect of this diabatic heating is most pronounced in the north of Greenland and the Canadian Arctic Archipelago, while the largest relative cooling at 1000 hPa within the Atlantic sector of the PASM occurs in the northern Barents Sea (Figs. 3b and S10e). The Barents Sea is a key region for releasing heat from the Atlantic Ocean, and Shu et al. (2021) noted that the interaction of sea-ice-atmosphere has allowed the "Atlantification" to extend to the northern Barents Sea. Under the background of global
warming, the subsurface Arctic warm blobs caused by "Atlantification" will become an important component of the positive feedback process of Arctic sea ice reduction (Voosen, 2020). Therefore, we speculate that the discrepancy between Fig. 5b and Fig. S10e is due to the influence of oceanic processes on sea ice. According to the Sverdrup theory of oceanic circulation, the curl of wind stress $\vec{\tau}$ is approximately equal to the product of the $\beta$ effect of planetary rotation and the water volume transport $V$, expressed as:

$$curl\,\vec{\tau} = \beta V \tag{14}$$

During the invasion of the ridge from the Atlantic sector, an anomalous anticyclone with high pressure is present southeast of Greenland (Fig. 6d). The negative vorticity of the anticyclone leads to an abnormal southward water volume transport $V$. This reduces the transport of warm and salty Atlantic water towards the pole, i.e., it weakens "Atlantification". Consequently, SIC increases in the central Arctic Atlantic sector, especially in the northern Barents Sea. Utilizing high-quality oceanic
observations in conjunction with models to explore the inference is one of our future research endeavors.

The amplitude of sea ice variability during the winter and spring seasons is inherently small and not easily observable, which necessitates more focus on sea ice changes while also posing challenges for data precision. Under current data conditions, our study focuses on the relative variations of Arctic sea ice, which highlight the commonalities of sea ice variability between the winter-spring and summer-fall periods. By selecting ELSEs based on daily data, the entire
development process of an extreme case may not be fully captured. Whatever, it effectively explores the spatial distribution



and commonalities in the causes of different types of ELSEs. This approach is of profound significance for sea ice prediction and simulation, and consequently for ecology, climate, and human society.

Previous studies have demonstrated the predominant role of thermodynamic processes in the variability of Arctic sea ice. Dumas et al. (2003) found that the variability and trend of Arctic sea ice thickness are primarily driven by
thermodynamics. Zhang et al. (2008) estimated that approximately 70% (30%) of sea ice loss in the Arctic Pacific sector is attributed to thermodynamic (dynamic) effects based on PIOMAS data. Olonscheck et al. (2019) indicated that atmospheric temperature fluctuations are the main driver of sea ice reduction, and that dynamic contributions accounting for about 25% of the sea ice variability. Spreen et al. (2020) showed that despite a decrease in Arctic sea ice export between 1992 and 2014, the overall volume of Arctic sea ice continued to decline, reflecting the dominant influence of thermodynamic processes in
Arctic sea ice reduction. Polyakov et al. (2022) highlighted that thermodynamic processes play a decisive role in the seasonal evolution and predictability of mean sea ice thickness, and that sea ice advection increasingly affects the predictability of seasonal sea ice. With the intensification of global warming and the rapid warming of the Arctic, we hypothesized that the thermodynamically dominated EWSM and PASM will occur more frequently. Furthermore, not all ELSEs are dominated by local thermodynamic processes. The unpresented EOF3 and EOF4 in this study are primarily related to anomalous
temperature advection from mid-latitudes, with significant increases in dynamic contributions. Their atmospheric circulation characteristics are markedly different from those of EOF1 (the EWSM) and EOF2 (the PASM), as will be discussed in future work.

*Data availability statement.* The National Snow and Ice Data Center (NSIDC) provides sea ice concentration (https://nsidc.org/data/nsidc-0051/versions/2) (DiGirolamo et al., 2022), sea ice motion (https://nsidc.org/data/nsidc-
0116/versions/4) (Tschudi et al., 2019a) and sea ice age (https://nsidc.org/data/nsidc-0611/versions/4) (Tschudi et al., 2019b) data. The Pan-Arctic Ice Ocean Modeling and Assimilation System (PIOMAS) offers sea ice reanalysis data (https://psc.apl.uw.edu/research/projects/arctic-sea-ice-volume-anomaly/data/model_grid) (Zhang and Rothrock, 2003). The ERA5 reanalysis data (https://cds.climate.copernicus.eu/cdsapp#!/dataset/reanalysis-era5-pressure-levels?tab=form, https://cds.climate.copernicus.eu/cdsapp#!/dataset/reanalysis-era5-single-levels?tab=form) (Hersbach et al., 2023a, b) are
provided by the European Centre for Medium-Range Weather Forecasts (ECMWF). The Japanese Meteorological Agency (JMA) offers the Japanese 55-year reanalysis (JRA-55) data, including large-scale condensation heating rate, convective heating rate, vertical diffusion heating rate, solar radiation heating rate, and longwave radiation heating rate (https://rda.ucar.edu/datasets/ds628.0/dataaccess/#) (Kobayashi et al., 2015).

*Author contributions.* All authors contributed to the conceptualization. Zhenlin Li and Fei Huang were responsible for data
curation, methodology development, and formal analysis. Zhenlin Li and Ruichang Ding performed code development and data visualization. All co-authors participated in result assessment. Zhenlin Li and Jian Shi drafted the original manuscript.



Fei Huang and Zhenlin Li managed funding acquisition, project administration, and supervision. All co-authors contributed to manuscript review and editing.

*Competing interests.* The authors declare that they have no conflict of interest.

*Acknowledgments.* We acknowledge the financial support by the National Key Research and Development Program of China, National Natural Science Foundation of China, and Fundamental Research Funds for the Central Universities.

*Financial support.* This work was supported by the National Key Research and Development Program of China (2019YFA0607004), National Natural Science Foundation of China (NSFC) Projects (42075024, 42430411), and Fundamental Research Funds for the Central Universities (202561001).

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
