# Peer review of "Thermodynamic and dynamic drivers underlying extreme central Arctic sea ice loss"

_EGUsphere, 2025_

## Referee Comment (RC2)

**Review for "Thermodynamic and dynamic drivers underlying extreme central Arctic sea ice loss", by Zhenlin Li, Fei Huang, Jian Shi, Ruichang Ding and Shumeng Zhang, manuscript submitted to The Cryosphere.**

December 9, 2025

**General comments**

The study conducted by Li *et alii* investigates events of major sea ice concentration decrease in the Central Arctic. To do so, they use an Empirical Orthogonal Function decomposition of standardized sea ice concentration (SIC) anomalies, and use a set of complementary analyses, sea ice budget, temperature tendency analysis, sea ice dynamic decomposition and Rossby wave source identification, to determine the drivers of the two dominant modes of the SIC anomalies. The topic is of interest, though already extensively studied, and the study could contribute to the overall understanding of how sea ice in the Arctic evolves, in particular for the next few decades, during which the Central Arctic sea ice will become more and more vulnerable. Moreover, the large range of complementary analyses to determine the sources of sea ice loss are at first read enticing.

Unfortunately, the methods at the heart of those analyses and the data used to conduct them raise some serious concerns and I therefore have doubts as to whether they can support the claims from the authors. But more importantly, the study never addresses the potential role of the ocean to drive the ELSEs, while it is now considered as the first driver of sea ice loss. Moreover, it does not seem to disentangle the trend from the EOF modes, while it is likely hidden in the two dominant modes.

I do my best to detail my concerns below and to support them convincingly. I suspect addressing them properly will require a significant amount of work, including a total change of the methodology used. It should also lead to a significant change in the results.

**Major comments**

In a nutshell:

- Using reanalysis data to close the temperature and sea ice budget should not be done

- The temperature tendency equation does not include the heat flux coming from "below" (sea ice or ocean), while this can be the first order driver

- The ocean is never considered in the drivers of sea ice loss

- Sea ice concentration trends are not removed from the anomalies before conducting the EOF decomposition and are not discussed neither, which is a clear lack

- There might be some important between sea ice divergence and velocity divergence

- Many of the suggested causalities are not supported by the results and could be the other way around

- Many important methodological precisions are missing

Two major concerns are related to the methods and data used in the budgets (Methods sections 2.4 and 2.5 and Results sections 4 and 5).

**Reanalysis data fluxes should not be used to close a budget**

An important caveat of reanalysis products is that they are not physically consistent. Indeed, when assimilating observational data into the model state, spurious fluxes are introduced, breaking the conservation of some properties, including momentum and mass: in reanalysis, "the system state estimate can undergo jumps, implying implicit non-physical sources, and rendering very difficult the physical interpretation of the time-evolving state. Methods have been employed to smooth out the discontinuities over finite times, but still leaving artificial imbalances in the solution." (Wunsch & Heimbach, 2007). While reanalysis products provide the best estimate of the state of the climate, they should not be used to calculate budgets, as they cannot physically close them. A better, physically-consistent alternative to reanalysis would be State Estimates products, but those are costly to compute (e.g. ASTE, Nguyen et al., 2021) and typically not available for the kind of investigations conducted here.

Unfortunately, this study relies on reanalysis products, ERA5 and JRA55 to calculate a temperature budget, and PIOMAS to close a sea ice budget. This is a major issue, especially considering that the most important terms of the budgets (diabatic heating for the temperature; thermodynamics for the sea ice) are calculated by making the assumption that those

budget are closed and that the residuals therefore correspond to the wanted term. Uncertainties are difficult to evaluate in reanalyses, and so it remains unknown whether using those data while significantly alter the results on the spatial and temporal scales considered here. But in doubts, I believe we have to make the assumption that the unphysical flux produced by data assimilation might not be negligible. Therefore, the method proposed here to evaluate the thermodynamical and dynamical contributions to sea ice loss is not sound. Note that this is a bit less worrisome for the sea ice budget, as the dynamical term is actually estimated by observations and not a reanalysis, but the thermodynamical term should still include not only the "real" thermodynamics but also a (hopefully small) spurious term related to the correction of the sea ice thickness by assimilation of observations into the PIOMAS $H_{eff}$.

**The Temperature tendency does not account for the ocean or sea ice**

Equation (1) in section 2.4 equates the temperature tendency to the advection, the adiabatic heating and the diabatic. The tendency, the advection and the adiabatic heating are computed using the ERA5 reanalysis (see above for a major caveat of using this data for a budget). The last term, arguably the most important (I regret the authors did not show the comparison of all the terms), is estimated by considering that it is equal to the residuals of the budget. This could be true if 1. the reanalysis could be used to close the budget (I have argued above that it is not the case) and 2. if it was the only term missing. Unfortunately, I believe that the heat flux at the surface is not accounted for, in this equation. Indeed, sea ice or ocean are important heat sources or sinks and are therefore likely to provide an important heat flux. In equation (1), it is implicitly in the residual, but the text describing the equation makes me think that the authors are not aware of it: "The diabatic heating rate can also be directly estimated by summing the large-scale condensation heating rate, convective heating rate, vertical diffusion heating rate *[this would be the sensible heat flux between atmosphere and ocean/sea ice]*, solar radiation heating rate *[did the authors account for upward solar radiation proportional to the albedo of the surface? it seems not]*, and longwave radiation heating rate *[another heat flux for which sea ice or ocean need to be accounted for, but with no mention of it in the text]* based on the JRA-55 datasets." This is particularly worrisome as this study focuses on sea ice, but the equation is never used to link temperature tendency to sea ice! And in some cases (including during ELSEs), we can expect this flux to be the first order driver of the temperature tendency. Therefore this budget is not closed and, unless I missed something fundamental in the methods, what the authors consider as the diabatic heating is actually not the diabatic heating alone.

Because of those two major concerns, the results described in this study cannot be fully trusted. Many of those results are overall consistent with the scientific literature (e.g. the

dominating importance of the thermodynamics over the dynamics in the sea ice budget, Le Guern-Lepage & Tremblay, 2023 or the importance of the "diabatic term" in the temperature tendency, over the other terms). But some other results are a bit at odds, to the best of my knowledge, e.g. the prominent importance of latent heat flux, which is rather supposed to be one or two orders of magnitude smaller than radiative and sensible heat fluxes (note that this could actually be related to another methodological issue, see last major comment).

**What is the role of the ocean in the sea ice loss?**

This says it all. The ocean is a complete blind-spot of this study, while it now explains over half of the sea ice melt in the Central Arctic (e.g. Carmack et al., 2015, Oldenburg et al. 2024).

**Are trends of sea ice concentration included in the EOF modes?**

In the preprocessing steps before decomposing the sea ice concentration into EOFs, the sea ice anomalies are computed by removing the climatology. But no trend seems to be removed. Considering the major changes that the sea ice is undergoing in the Arctic, I would expect the trend to be the dominant mode of the EOF decomposition. The authors first briefly mention that indeed the first mode of the non-normalized anomalies "spatially manifest as significant SIC anomaly signals along the Arctic marginal seas and the edge of the central Arctic" (l. 220). The authors claims this is due to the summer signal; my guess is that this should also include the overall trend. The authors then normalize the anomalies to give equal weights to other seasons. But I would not expect this normalization to remove the trend. Yet, I am surprised to not see any mention of it when analysing the EOF decomposition of the standardized anomalies. Is that because it only appears in the third or higher order mode? Or because the method does indeed remove the trend? Or because the trend is actually not a major mode of variability? If the latter, this would be a major result that should be discussed. If not, I suspect it should be hidden somewhere and needs to be analysed and discussed. Moreover, in general, EOF decomposition studies tend to first remove the trend. I believe this needs to be done here as well. Note that this is not straight-forward for sea ice concentration, as this typically leads to sea ice concentrations above 1 at the beginning of the period of interest, and that a trend needs to be computed for each day-of-year (e.g. Richaud et al., 2025 for an example of day-of-year trend calculations for atmospheric variables).

**Sea ice divergence is not the same as Helmholtz divergent term**

In section 2.5, the sea ice dynamical term is decomposed into a advective and diverging term (eq. 3). Then in section 2.6, an Helmholtz decomposition of the (sea ice) velocity field is done, computing the divergent and rotational term. The text leads to think that the diverging term of eq. 3 and that of eq. 7 are equivalent. This is not the case and was (still is) very confusing to me. Those kinds of Helmholtz decompositions are typically done in rheological studies, but this is not the case here. Moreover, the text gives the impression that since dynamics can be decomposed into advection and divergence, and since the velocity field can be decomposed into divergence and vorticity, then the advection is equivalent to the vorticity: "By decomposing the standardized SIM fields into divergence (Fig. 9a and b) and vorticity (Fig. 9c and d) components, it can be observed that the sea ice advection primarily drives the anomalies in sea ice motion" (l. 312). This is obviously not the case. I do not understand why the advection was not directly computed, or if it was, why it wasn't shown and relied upon, rather than going through the rotational/vorticity. In any case, the Helmholtz decomposition does not bring anything to the study and I would suggest to drop it.

**Many of the assertions are not supported by the analyses**

"The above analysis suggests that local surface temperature anomalies caused by local diabatic heating anomalies are the important factors in the formation of EWSM and PASM." (l.243): I do not see how this sentence is supported. It suggests that atmospheric (surface) temperatures drive the two EOF modes found in the study, on the basis that the spatial patterns of the composite temperature match the EOF mode patterns. But it could very well be (and I would guess likely is) the opposite, with the ice pattern driving the temperature. This is one example, amongst many, of a causal link claim made by the authors that could very well be the other way around. And that reversed causality is never explored or mentioned. Other examples include l. 285-286, l. 317-318, l.330-332, l.341-342, l.343-345 (list not exhaustive). Moreover, some other claims in the Conclusions and Discussions section do not seem to be really demonstrated in the paper: "Under thermodynamic dominance, both the EWSM and the PASM trigger water vapor and cloud feedbacks to sustain and enhance their development" (l.386-387). Section 4.2 does discuss this but does not provide any result to prove it and Figure 7 just gives some vague (not convincing to me) spatial coherence between the different metrics. Same with l.394: "the convergence and divergence of SIM exert a minor positive contribution to the EWSM": I could not find any substantial result in the main text that support this.

Proving the causal link is complex, requires using some causality methods (e.g. Liang-Kleeman), and seems outside the scope of the study. Nonetheless, the claims of the authors

are a bit too assertive to my opinion, and a more nuanced view on the direction of the links needs to be taken into account.

**Many important aspects of the methodology are missing**

The description of the methodology at the moment does not allow to reproduce the results. For example, the temperature tendency description (section 2.4) does not mention if this is for surface (2m) temperature, atmosphere-integrated temperature), boundary layer temperature or else. See also above for the lack of description of the trends of sea ice concentration, and other variables as well, if any trend is accounted for. The calculation of the climatology is not sufficiently described (see also minor comments on a suggestion to smooth it). None of the units of the terms are ever given (if they had been, it would have become obvious that the different "divergent" terms are not the same). The temporal threshold for the detection of ELSEs is not given. Looking at the figures, there might not be any, which then is a potential point of improvement of the study (see minor comment). Finally, references are missing for nearly all important equations used in the method.

**Minor comments**

- Many equations are not referenced, such as the temperature tendency equation, the Helmholtz decomposition, the diabatic heating rate calculation, etc. I know those are classic equations, but they can take alternative forms depending on the field of interest, and therefore a quick reference towards other papers using those equations in the same way would be relevant.

- Sea ice observation data: why only start in 1989? Sea ice concentration and motion data are available starting in 1979, which would give another decade of precious data on an else relatively short time series. This would give a more robust analysis.

- Climatology calculation: the methods are not very explicit on the way the climatological mean is computed. It seems to be simply the mean of each day of the year, I suspect over the whole time series. A justified choice of the baseline would be good: 1979-2007 would avoid the recent decline period; or on the contrary only take the last 30 years to have the most recent behaviour, or the whole period? See Smith et al. (2025) for an in-depth discussion of why baseline are important. On top of that baseline aspect, it is conventional to smooth out the climatology when using daily data, by using a window around the considered day-of-year, yielding the advantage of increasing the sample size (see the MHW field of research, e.g. Hobday et al., 2016). This does not seem to have been the case in this study, when looking at Fig. 1.d).

- ELSE definition: there does not seem to be any temporal threshold on the detection of the ELSEs. In other word, a sea ice concentration below the 1.5 standard deviation threshold for 1 day would count as an ELSE, as would an event lasting a year. No discussion is brought on that aspect, and it seems to me that this requires some thinking and a different definition could yield different results. Considering the different temporal scales of the atmosphere, the ocean and the sea ice, the choice of the temporal threshold would give more or less weight on events that are likely to influence larger scale dynamics. I would recommend to filter out events shorter than a specific threshold, to be justified (e.g. 10 days? 1 month?). This would likely change the results in section 3.3: why is the 2008 event included as a significant one for the PASM mode, but not 2003?

- On the same aspect of ELSEs definition, I was very surprised to see that 2012 is not included in the list of ELSEs that are related to the EWSM and PASM modes, while it is the observational record of sea ice low. Considering its spatial pattern, I would expect it to be maybe in a PASM positive phase, but also with some EWSM negative contribution (wild, vaguely educated guess ;)). That does not seem to match Fig. 2, and I am curious as to why it is not in phase opposition with 2007 and why it is not more prominent. The study should at the very least discuss this aspect, considering the importance of 2012 in the recent sea ice evolution. Similarly, 2007 is also a very important sea ice low event: a couple of sentences on how it fits in the EOF modes story would be valuable.

- The text only mentions the first two modes, which seem to explain 33 % of the total variance. This means the other modes are also important. It seems that the authors would like to discuss the other modes in another study, but I think it would be important to at least mention how quickly the explained variance decreases with the other modes. Moreover, the fact that the first two modes explain "only" 33 % of the total variance of sea ice concentration standardized anomalies seem to indicate that the EOF decomposition might not be the best approach to explain the variability of sea ice. A discussion on this aspect seems important.

- Figure S7 shows the composite differences of the standardized anomalies for the different heat fluxes. First, decomposing the radiation flux into solar (shortwave) and thermal (longwave) radiation would be valuable. Second, no description of how those composite differences and anomalies are computed, leaving the reader guessing the methodology. Is that the standardized anomalies of each flux? Or is it the composite anomalies of the fluxes but for the EOF modes of the standardized anomalies (of sea ice)? No units, no labels are given on the colorbar to help me decide. But I suspect it is the first option,

looking at the colorbar values. If that's the case, it means each flux is normalized by its own standard deviation. But then we certainly cannot compare those fluxes together! The absolute value of the latent heat flux could be (and likely is, to the best of my knowledge) orders of magnitude smaller than the absolute value of the radiative fluxes: we have no information on that aspect in this manuscript. If that is the case, the claim that "Fig. S7 indicates that latent heat flux anomalies are prominent in the formation of the EWSM and PASM" (l.249-250) could be wrong, unfortunately.

- On a related topic, the reader misses critical information to understand how the SIM anomalies are standardized: are U&V standardized by the total velocity SD, or by U SD for U and by V SD for V? I suspect the first option, as the second would not make any sense and would prevent any comparison, but some weird patterns on Fig. 8 (e.g. on the Siberian Shelves for 8.b) cannot take off my mind that the second option might be used. Please provide the necessary details.

- Divergence: "There is an out-of-phase relative variation of D between the Eastern and Western Hemispheres composited for EWSM mode" (l. 315). As already mentioned in the major comments, I am very confused by the use of sea ice divergence in the sea ice (thickness) budget and in the Helmholtz decomposition; there is also a third "divergence" used in this study, defined by eq. 6 and written DF. This is not the same divergence as the other two, since it uses SIC, instead of $H_{eff}$ for the sea ice budget divergence and no sea ice for the Helmholtz decomposition. Yet, it does not seem to be distinguished in this section. I suspect that the DF and the ice thickness divergent term might be similar, but they would still show some differences.

- The whole section 6 on the Rossby wave source and teleconnection seems out of place. It does not use the same approach (why not use the composite differences as done for all other aspects?), it is not clearly linked to the EOF modes, Fig. 11 only shows the Pacific side of the Arctic (why not the rest) and its contribution to new scientific knowledge is not obvious to me (I believe the Pacific role in the Rossby wave generation is well known and that the spatial pattern of the T-N wave activity flux has also already been documented extensively). Therefore, I do not understand the contribution of this section to the general scientific knowledge. But I admit that this is a bit far from my domain of expertise and I might simply not be knowledgeable enough to understand its significance. I let the other reviewer(s), the editor and the authors judge on that aspect.

- Oceanic discussion (l. 413-435): This is the first and only time that the ocean is considered in the story. Unfortunately, the Sverdrup balance is not valid for the Arctic (e.g. Timmermans & Marshall, 2020), and the inflow of warm Atlantic water is governed

by a complex set of atmospheric and oceanic interactions. Moreover, the Sverdrup balance is brought up in an attempt to explain discrepancies between Fig. 5b and S10e. The issue is that those are not showing the same thing at all! Fig. S10e shows surface temperature anomalies (in K) while Fig. 5b shows the diabatic rate (which should be in K/s); comparing that latter figure to the temperature tendency anomalies could be done, but not to the actual temperature anomalies. Hence, we should not expect a similarity between Fig 5b and S10e. Regarding the oceanic heat transport in the Arctic, a good source of information are Docquier & Koenigk (2021) and Docquier et al. (2021). Check also Polyakov et al. (2023). Many other papers investigate the role of heat inflow into the Arctic and would be a much more reliable and convincing source of explanation of Atlantification than the proposed hand-wavy Sverdrup balance that is not applicable.

**Suggestions, technical details and typos**

**Text**

- Use of "vorticity" (e.g. l. 313), "rotational" (e.g. l. 154) and "nondivergent" (e.g. Fig. 9) names for the same term: please pick one term and stick to it. Same for divergent vs. irrotational.

- l.10: remove "First,"

- l.14: "which highlight common characteristics of sea ice variations." We could argue that this is not really the case, and that concentration anomalies are simply a convenient metric to observe, but that ice thickness would be much better to really understand sea ice variation.

- l. 28, "approximately twice the global average": the most recent estimates rather suggest 3 to 4 times (e.g. Rantanen et al. 2022)

- l. 33, "This rapid warming in the Arctic has led to a significant decline trend in Arctic sea ice ": one could argue that the decline in Arctic sea ice has led to the rapid warming, more than the opposite (albedo positive feedback). Please nuance this sentence.

- l. 47: Voosen (2020) seems like a journalistic piece, not a scientific article. While it is an interesting one, please provide a peer-reviewed paper. Moreover, I could not find any part of that (short) piece that talks about a dynamic and thermodynamic coupling... was that LLM-generated?

- l. 51: "Arctic sea ice concentration" or "extent"

- l. 54: Sticker et al. (2025) would be a good, recent addition here

- l. 61: Hoffman et al. (2025) would be a good, recent addition here

- l.63, " the reduction of Arctic sea ice is spatially heterogeneous, which is attributable to the spatial variation of thermodynamic and dynamic processes driven by atmospheric and oceanic circulation. [...] The trend of sea ice reduction is notably significant in the marginal seas along the Eurasian and the North American coast, while the sea ice from north of Greenland and Canadian Arctic Archipelago to the pole remains relatively stable". I agree that atmospheric and oceanic circulation play an important role in generating spatial variability, but not those described by the text. The difference between shelves and northern part is simply due to astronomical considerations (less solar radiation close to the pole than further south, e.g. Maksym 2019)... Please rephrase.

- l.71, "the perennial sea ice in the central Arctic has begun to undergo extreme reductions in recent years": "recent" is subjective, but it has been a few decades by now, so I would remove the "begun [...] in recent years"

- l.76, "This is because the sea ice in the central Arctic region is predominantly multi-year thick ice, and the absolute value of sea ice variation in winter and spring is relatively small". It is also (and maybe primarily) because the winter and spring sea ice extent is strongly geographically constrained by the surrounding continents, and that there is therefore less degree of freedom (e.g. Maksym, 2019). Please rephrase.

- l. 179-186: This is a good introduction, though maybe a bit too detailed. You could shorten to only highlight the relevant definition.

- l.187, "climatological SIC": is that annual mean? seasonal? day-of-year? Maybe worth considering a time-varying definition. See also minor comment for smoothing suggestion, to have a more statistically robust definition. If it is annual (as Fig 1 seems to suggest), it needs some discussion, as it will induce a significant ELSE detection bias between winter and summer.

- l.193, "which validates the rationality of the definition of the central Arctic in this study": except for JAS, for which the probability of SIC¿90 % represents a small fraction of the Central Arctic. See above suggestion to use a time-varying definition.

- l.225-226: please provide references for those EWSM and PASM: are those names yours? Or do they come from other studies? Do they match other studies?

- l. 241: how is the spatial correlation computed? Why is it only computed for air temperature and not for the diabatic heating, heat fluxes, etc? Also, see major comments: correlation is not causation.

- l. 334, "Considering that the most pronounced EWSM signal occurs during the winter of 2002 [...]"

- l. 454-455, " The unpresented EOF3 and EOF4 in this study are primarily related to anomalous temperature advection from mid-latitudes, with significant increases in dynamic contributions.": I don't really understand the distinction made between dynamic and thermodynamic, then. To me, the advection of heat would lead to a change in thermodynamics, not dynamics, which would rather be controlled by momentum fluxes, not heat fluxes. So this sentence seems self-contracting to me.

**Figures**

- All: please provide labels + units on all colorbars and don't hesitate to also add title above the different panels to make sure the reader can quickly understand what they are looking at. Try to be consistent and homogeneous between figures, keeping the same latitude boundaries, projections, row/column orientation, etc.

- Fig. 1: The colormap for panels a and b is not sequential, consider using another one; please mask regions where there is no ice, instead of plotting the 0% SIC. Panel e (and left panels in figure 2) are great! I would suggest making those a bit wider to see better.

- Fig. 3: this figure is a bit confusing because at first glance, it is not clear that adding advection and divergence leads to the Dynamics term. Please consider stacking them to reducing their width and adding transparency to make it more obvious that there are only two terms and that you decompose one of them into two.

- Figures 6 and 9: I would suggest to transpose the figures, putting the EWSM and PASM as rows instead of columns, to match the other figures (e.g. Fig 7 and those in SI). Keeping the same convention would allow the reader to be able to skim through and compare the figures in a more intuitive way.

- Figs. 10 and 11: Why change the projection? Other figures use a North Polar Stereo projection while 10 and 11 are cylindrical (?).

- Fig. 11: Why not plotting the whole hemisphere? At least for a) and d).

- Fig. 12: Great schematic! Not sure it is very colorblind-friendly, but I like it. I am unfortunately not convinced by the content, because of all the reasons detailed in the major comments...

**References**

- Carmack, E., Polyakov, I., Padman, L., Fer, I., Hunke, E., Hutchings, J., Jackson, J., Kelley, D., Kwok, R., Layton, C., Melling, H., Perovich, D., Persson, O., Ruddick, B., Timmermans, M.-L., Toole, J., Ross, T., Vavrus, S., and Winsor, P.: Toward Quantifying the Increasing Role of Oceanic Heat in Sea Ice Loss in the New Arctic, Bulletin of the American Meteorological Society, 96, 2079–2105, `https://doi.org/10.1175/BAMS-D-13-00177.1`, 2015.

- Docquier, D. and Koenigk, T.: A review of interactions between ocean heat transport and Arctic sea ice, Environ. Res. Lett., 16, 123002, https://doi.org/10.1088/1748-9326/ac30be, 2021.

- Docquier, D., Koenigk, T., Fuentes-Franco, R., Karami, M. P., and Ruprich-Robert, Y.: Impact of ocean heat transport on the Arctic sea-ice decline: a model study with EC-Earth3, Clim Dyn, 56, 1407–1432, `https://doi.org/10.1007/s00382-020-05540-8`, 2021.

- Hobday, A. J., Alexander, L. V., Perkins, S. E., Smale, D. A., Straub, S. C., Oliver, E. C. J., Benthuysen, J. A., Burrows, M. T., Donat, M. G., Feng, M., Holbrook, N. J., Moore, P. J., Scannell, H. A., Sen Gupta, A., and Wernberg, T.: A hierarchical approach to defining marine heatwaves, Progress in Oceanography, 141, 227–238, `https://doi.org/10.1016/j.pocean.2015.12.014`, 2016.

- Hoffman, L., Massonnet, F., and Sticker, A.: Probabilistic Forecasts of September Arctic Sea Ice Extent at the Interannual Timescale With Data-Driven Statistical Models, Journal of Geophysical Research: Machine Learning and Computation, 2, e2025JH000669, `https://doi.org/10.1029/2025JH000669`, 2025.

- Le Guern-Lepage, A. and Tremblay, B. L.: Disentangling Dynamic from Thermodynamic Summer Ice Area Loss from Observations (1979–2021): A Potential Mechanism for a "First-Time" Ice-Free Arctic, Journal of Climate, 36, 7693–7713, `https://doi.org/10.1175/JCLI-D-22-0628.1`, 2023.

- Maksym, T.: Arctic and Antarctic Sea Ice Change: Contrasts, Commonalities, and Causes, Annual Review of Marine Science, 11, 187–213, `https://doi.org/10.1146/annurev-marine-010816-060610`, 2019.

- Nguyen, A. T., Pillar, H., Ocaña, V., Bigdeli, A., Smith, T. A., & Heimbach, P. (2021). The Arctic Subpolar gyre sTate Estimate: Description and assessment of a data-constrained, dynamically consistent ocean-sea ice estimate for 2002–2017. Journal of Advances in Modeling Earth Systems, 13, e2020MS002398. `https://doi.org/10.1029/2020MS002398`

- Oldenburg, D., Kwon, Y.-O., Frankignoul, C., Danabasoglu, G., Yeager, S., and Kim, W. M.: The Respective Roles of Ocean Heat Transport and Surface Heat Fluxes in Driving Arctic Ocean Warming and Sea Ice Decline, Journal of Climate, 37, 1431–1448, https://doi.org/10.1175/JCLI-D-23-0399.1, 2024.

- Polyakov, I. V., Ingvaldsen, R. B., Pnyushkov, A. V., Bhatt, U. S., Francis, J. A., Janout, M., Kwok, R., and Skagseth, Ø.: Fluctuating Atlantic inflows modulate Arctic atlantification, Science, 381, 972–979, `https://doi.org/10.1126/science.adh5158`, 2023

- Richaud, B., Dowd, M., Renkl, C., and Oliver, E. C. J.: Sea Ice Nonlinearities Act to Rectify and Filter Oceanic and Atmospheric Forcing, Journal of Climate, 38, 4573–4588 `https://doi.org/10.1175/JCLI-D-24-0485.1`, 2025.

- Smith, K. E., Sen Gupta, A., Amaya, D., Benthuysen, J. A., Burrows, M. T., Capotondi, A., Filbee-Dexter, K., Frölicher, T. L., Hobday, A. J., Holbrook, N. J., Malan, N., Moore, P. J., Oliver, E. C. J., Richaud, B., Salcedo-Castro, J., Smale, D. A., Thomsen, M., and Wernberg, T.: Baseline matters: Challenges and implications of different marine heatwave baselines, Progress in Oceanography, 231, 103404, `https://doi.org/10.1016/j.pocean.2024.103404`, 2025.

- Sticker, A., Massonnet, F., Fichefet, T., DeRepentigny, P., Jahn, A., Docquier, D., Wyburn-Powell, C., Quint, D., Shivers, E., and Ortiz, M.: Seasonality and scenario dependence of rapid Arctic sea ice loss events in CMIP6 simulations, The Cryosphere, 19, 3259–3277, `https://doi.org/10.5194/tc-19-3259-2025`, 2025.

- Timmermans, M.-L. and Marshall, J.: Understanding Arctic Ocean Circulation: A Review of Ocean Dynamics in a Changing Climate, Journal of Geophysical Research: Oceans, 125, e2018JC014378, `https://doi.org/10.1029/2018JC014378`, 2020.

- Wunsch, C. and Heimbach, P.: Practical global oceanic state estimation, Physica D, 230, 197–208, 2007.

---

## Author Comment (AC1)

**Subject:**

Response to reviewer #1 for manuscript [egusphere-2025-3594: "Thermodynamic and dynamic drivers underlying extreme central Arctic sea ice loss"]

Text in black: Reviewer's comments; Text in blue: Authors' responses.

**Introduction**

This manuscript explores the changes to central Arctic sea ice concentration (SIC) by identifying episodes of intense SIC loss ELSEs or Extreme Loss Sea ice Events. This is done through a rare event threshold and is inferred from the NSIDC SIC record. Secondly, the manuscript performs Empirical Orthogonal Function analysis upon the SIC fields to identify two dominant modes of SIC anomaly. Finally, the authors budget the two modes of SIC using PIOMAS and explore the connections to atmospheric feedbacks. The paper is well written, thorough and the visual representations are well produced. However, there are some considerations to make before progressing with this manuscript, specifically with the clarity of methodology.

**Novelty**

The novelty in this paper is rooted in the identification of modes of SIC ELSEs. Namely, the East-West seesaw (EWSM) and Pacific-Atlantic seesaw (PASM) modes. These are interesting concepts for explaining the differing behaviours of the Arctic sea ice system during loss events. As well as the interpretation of atmospheric feedbacks that induce these modes.

We greatly appreciate your detailed review of this manuscript, which has significantly helped us improve the manuscript.

**Positioning**

The paper is well placed within the literature to move forward with identifying system changes to the central Arctic sea ice regime. However, there are several key papers within the field that are not cited. See references.

We thank you for the valuable references, which we will adopt and cite to enhance our paper.

The sea ice budget diagnostic method used in this study is closely related to Holland and Kwok (2012) and Holland and Kimura (2016). Their works have well quantified the contributions to sea ice variations in both the Arctic and Antarctic, laying the foundation for the sea ice budget diagnostic analysis. Massonnet et al. (2018) demonstrates that sea ice variations mainly depend on the mean state (primarily SIT) of sea ice, which emphasizes the crucial role of SIT in the sea ice budget analysis here. These references highlight the necessity of improving the elaboration of our methods. For specific details, refer to the italicized part of our response to the second question below. In addition, Massonnet et al. (2018) argues that Arctic sea-ice change is tied to its mean state mainly through thermodynamic processes. This supports the conclusion of this study that the first two modes of sea-ice changes in the central Arctic are mainly influenced by thermodynamic effects and we will present this connection in the revised manuscript.

**General Comments**

The authors should confirm the prevalence and trend of SIC ELSEs using an alternative SIC product, e.g. OSISAF-401. I imagine this to have minimal change in event distribution; however, the impact

upon EOF modes may be stronger. The work is robust in using multiple reanalysis datasets for atmospheric variables; the same approach should be taken concerning the SIC product.

We appreciate your constructive comments. Validating the robustness of SIC ELSEs and EOF modes is crucial to the research significance of this paper.

We note that the OSISAF-401 dataset, with a time span from 2005 to the present, does not cover the research period of this study and is thus inappropriate for use. The OSISAF-450a1 product spans 1978–2020, while OSISAF-430a serves as its supplementary dataset covering 2021–2025. The two products are suitable for this study, and numerous studies have used them for research about the Arctic (Belter et al., 2021; Docquier and Koenigk, 2021; Sumata et al., 2022, 2023; Tian et al., 2022). Therefore, we adopt OSISAF-450a1 and OSISAF-430a for validation in this study.

The spatial pattern of the central Arctic derived from OSISAF data is nearly identical to that from NSIDC, though the extent of the central Arctic in OSISAF data is slightly larger (Fig. R1a, b). The seasonal and interannual variability characteristics of sea ice in OSISAF data are consistent with those in NSIDC, but with smaller amplitudes (Fig. R1c, d). While the distribution of ELSEs between the two datasets differs in winters after 2015, both datasets effectively capture the key features of ELSEs (Fig. R1e): 1. The amplitude of ELSEs is stronger in summer than in winter; 2. 1990, 2007, 2012, 2016, and 2020 are typical years with ELSEs; 3. The frequency and intensity of ELSEs have increased significantly in recent years. In summary, the comparison between OSISAF and NSIDC SIC datasets indicates that the distribution and characteristics of ELSEs are robust.

Figure R1. The same as Fig. 1 in the manuscript but for OSISAF SIC product results.

When comparing the standardized SIC anomaly fields of ELSEs, the most notable feature is that the signal in winter 1990 from OSISAF data is significantly stronger than that from NSIDC, while the signal in November and December 2016 is relatively weaker (Fig. R2a). The spatiotemporal modes derived from EOF decomposition using OSISAF are nearly identical to those from NSIDC, with the caveat that the order of the first two modes is reversed (Fig. R2b, c). The PASM is prominent in 1990; thus, the stronger signal in 1990 from OSISAF data compared to NSIDC (Fig. R2a) increases the

variance contribution of PASM, making it the first EOF mode. Above all, despite slight differences in the central Arctic region, ELSEs distribution, and standardized SIC anomaly fields of ELSEs between the two datasets, both ultimately identify the EWSM and PASM, confirming the robustness of these two modes in the sea ice variability over the central Arctic.

Relevant contents and figures will be added to the revised version and supplementary information.

Figure R2. The same as Fig. 2 in the manuscript but for OSISAF SIC product results.

It is not obvious to me where the SIC budget is drawn from. Are the drift products used within the budget all from PIOMAS, or are they observational? If the budget is solely constructed from PIOMAS, how well does PIOMAS align with the observations, not only on the existence of ELSEs but also the distributions? If PIOMAS does not produce any ELSEs, how can the sea ice budget be well constrained or relevant for these modes? The whole methodological section on budgeting needs to make clear what is used and why.

Thanks for your reminder regarding the lack of necessary elaboration in the Methods section of the paper.

In the sea ice budget diagnosis, we used observed data of SIC and SIM (Sea Ice Motion) from NSIDC, which ensures the robustness of ELSEs in the diagnostic process. Since satellites cannot directly observe SIT (Sea Ice Thickness), NSIDC lacks SIT data with the spatiotemporal coverage and

resolution required for this study. Therefore, we adopted SIT data from PIOMAS. It is worth noting that the diagnostic analysis combining observational and reanalysis data introduces an equation closure issue. We again appreciate the references you provided; through these references and those citing them, we found that previous studies have analyzed and discussed the equation closure issue caused by different data sources, concluding that it is reliable to use NSIDC SIC and PIOMAS SIT data for sea ice budget diagnosis.

We will revise the manuscript based on the *following text* to enhance the clarity of the Methods section.

Previous studies proposed using SIC budget to diagnose sea ice changes, in which the advection term, divergence term, and residual term (including thermodynamic processes and mechanical processes related to ridging and rafting) constitute the total SIC budget (Holland and Kimura, 2016; Holland and Kwok, 2012). Schroeter et al. (2018) replaced the SIC budget with the sea ice volume budget, thereby enabling SIT changes to include mechanical processes like ridging and rafting and making the residual term directly represent thermodynamic processes; this method has been widely used (Bi et al., 2023; Ding et al., 2025; Lukovich et al., 2021).

The sea ice volume budget analysis requires SIC, SIM, and SIT data, among which SIC and SIM are derived from NSIDC observational data in this work to ensure the robustness of ELSEs in the diagnostic process. Previous studies have indicated that the simulation bias of SIT dominates the accuracy of Arctic sea ice simulation and prediction (Massonnet et al., 2018). Therefore, SIT data is crucial for the diagnostic analysis of sea ice budget. Given that SIT cannot be directly observed by satellites, we use the SIT reanalysis data from PIOMAS here.

It should be noted that the combination of observational and reanalysis data for budget diagnosis involves the equation closure issue. Previous studies proposed the Lagrange multiplier algorithm to address the issue (Mayer et al., 2018). Existing research has used this method to conduct sea ice budget analysis based on NSIDC and PIOMAS data (Ding et al., 2025; Lukovich et al., 2021). They found that the difference between the original budget terms and those corrected by the Lagrange multiplier algorithm is negligible, indicating the sea ice budget diagnostic method combining NSIDC and PIOMAS data here is reliable.

There is a clear change in the behaviour of ELSEs of the PASM in Figure 2c. Naturally, there should be a discussion about the shifting seasonality of this phenomenon that the work is missing.

Thank you for your academically insightful comment; this is a very interesting phenomenon.

As shown in Fig. 2c, the PASM mainly exhibits negative phases in winter and positive phases in spring and summer. The most typical negative phase events occurred in 1990, 2002, and 2016; the most typical positive phase events occurred in 2007, 2012, and 2016. To our knowledge, the surface layer of the Arctic Atlantic sector was dominated by a strong low-pressure system in the winter and spring of 1990, while a significant low-pressure system was present in the Pacific sector during the summer of 2012. These low-pressure systems are conducive to the occurrence and maintenance of Arctic cyclones. Considering that the thermodynamic and dynamic coupling effects of Arctic cyclones play an important role in forcing sea ice reduction (Aue et al., 2022; Cavallo et al., 2025; Finocchio et al., 2020, 2022; Lukovich et al., 2021), we hypothesize that Arctic cyclones are a key factor contributing to the seasonal phase transition of the PASM.

Based on the cyclone identification and tracking algorithms developed by Zhang et al. (2004) and Crawford and Serreze (2016), we created an Arctic cyclone dataset using ERA5 sea level pressure data.

We found that in the central Arctic, super cyclones during typical negative-phase PASM events only occur in the Atlantic sector (Fig. R3d–f), whereas during typical positive-phase PASM events, the Pacific sector is also affected by super cyclones (Fig. R3a–c). Furthermore, compared with the negative phase, the positive phase of PASM is characterized by a significant decrease in the probability density of super cyclones in the Atlantic sector and a significant increase in the Pacific sector (Fig. R3g); similarly, from winter to summer, the probability density of super cyclones decreases in the Atlantic sector and increases in the Pacific sector (Fig. R3h). Previous studies have conducted numerous case analyses on the impact of super cyclones on Arctic sea ice reduction (Lukovich et al., 2021; Stern et al., 2020; Tian et al., 2022). Our findings are consistent with these cases in terms of temporal distribution and the spatial pattern of sea ice reduction. These results fully indicate that the seasonal phase transition of PASM is closely linked to super cyclones.

We will add relevant content in the revised version to discuss this issue, thereby enriching the understanding of PASM.

**Figure R3.** Sea level pressure anomalies (shadings) and distribution of super cyclones (central pressure >1.5 SDs below the climatological monthly mean; purple dots) during typical positive-phase (a-c) and negative-phase (d-f) PASM events. Differences in super cyclone probability density between typical positive and negative phases of PASM (g) and between winter and summer (h). Magenta lines represent the central Arctic.

The brief section on Arctic amplification in the introduction is well read but needs to be restructured for clarity.

We will revise "Amidst the context of global warming, the Arctic experiences a greater warming rate that is approximately twice the global average, a phenomenon commonly referred to as Arctic Amplification (Pithan and Mauritsen, 2014; Serreze and Barry, 2011). Recent research has highlighted that Arctic Amplification has been underestimated (Chylek et al., 2022; Rantanen et al., 2022). Both observations and modeling studies have revealed that the Arctic region is warming at a rate approximately four times faster than the global average, with the potential to surpass eight times that of the global average in winter (Davy and Griewank, 2023)." to "Amidst the background of global warming, the Arctic is warming at a rate roughly twice the global average—a phenomenon widely known as Arctic Amplification (Pithan and Mauritsen, 2014; Serreze and Barry, 2011). However, recent studies have indicated that this amplification has been underestimated: both observational and modeling studies now reveal that the Arctic is warming at a rate about four times (even potentially exceeding eight times during winter) faster than the global average (Chylek et al., 2022; Davy and Griewank, 2023; Rantanen et al., 2022)."

We have improved the logical coherence of this paragraph by incorporating dashes, colons, the conjunction "however", and temporal markers such as "now". We aim to present the evolving understanding of the Arctic Amplification phenomenon concisely and clearly through such expressions. We would greatly appreciate any further suggestions if possible.

**Technical Comments:**

L27-28: Remove "a greater" in the sentence on Arctic Amplification

L35: "Significant declining trend" not a "significant decline trend"

L38-39: "acts as a pivotal modulator to drive the evolution" should be "acts as a pivotal modulator of the evolution"

L43: "What's more" is too informal

L119: "Cases when the principal components" should be "Cases in which ..."

L290: remove "directly"

L440: "Whatever" should be However or Nevertheless

We sincerely appreciate your thorough revisions, which have significantly improved the manuscript's language. We will incorporate all suggestions in the revised version.

**Figures/Tables:**

Figure 11 – either the stippling needs to change to a more appropriate and eye-catching colour, or the colourmaps. The cyan is lost in 11c/11f in the low advection regions

Thank you for your suggestion. We have revised Fig. 11 to ensure clear presentation of information (Fig. R4).

Figure R4. The same as Fig. 11 but replace the cyan stippling with magenta crossings.

Figure S11 – is a very strong plot demonstrating the modal deformation anomalies. What is the equivalent plot for EWSM?

The result can be found in Fig. R5a. For the EWSM, the sea ice deformation anomalies are almost entirely positive across the Arctic Ocean, a phenomenon that is interesting and worthy of discussion. Here, we propose a potential physical explanation. In the Canadian Arctic Archipelago and northern Greenland, increased sea ice deformation facilitates the significant outward advection of thick ice in the EWSM as mentioned in our manuscript. As this thick ice is transported toward the marginal seas of the Eastern Hemisphere, it melts and thins, with deformation intensifying simultaneously. This process also promotes the spreading of thick ice into a more uniform distribution, leading to increased sea ice concentration. In brief, intensified sea ice deformation in the thick ice regions of the Western Hemisphere within the central Arctic is one of the causes of EWSM formation, whereas intensified deformation in the Eastern Hemisphere is a consequence during the EWSM development process.

Furthermore, the impact of sea ice deformation on the EWSM and PASM differs in magnitude. The EWSM is more pronounced in winter than in summer as evident in Fig. 2b. Thus, even if intensified sea ice deformation leads to increased open water, the weak solar radiation in winter has a negligible effect on sea ice melting. In contrast, the PASM is significantly more influenced by sea ice deformation. As shown in Fig. 2c, positive phases of the PASM are concentrated in summer; intense sea ice deformation in the Pacific sector (Fig. R5b) would increase open water, enhancing oceanic absorption of solar radiation and thereby accelerating sea ice melting. Notably, the intense sea ice deformation in the Pacific sector during the PASM is closely linked to the dynamic effects of super cyclones (Fig. R3), which further corroborates our

earlier discussion on the influence of super cyclones on the seasonal phase transition of the PASM.

Initially we did not present the result of Fig. R5a in the manuscript considering the above. We believe that sea ice deformation anomalies are not solely the cause of EWSM formation but also its consequence, which requires more detailed hemispheric-specific research. To minimize potential confusion for readers, we will include a brief discussion of this issue in the revised manuscript.

**Figure R5.** Composite differences between the positive and negative phases of the standardized anomalies of sea ice deformation for the (a) EWSM and (b) PASM.

**References**

Holland, P. R., & Kimura, N. (2016). Observed concentration budgets of Arctic and Antarctic sea ice. *Journal of Climate*, 29(14), 5241-5249.

Massonnet, F., Vancoppenolle, M., Goosse, H., Docquier, D., Fichefet, T., & Blanchard-Wrigglesworth, E. (2018). Arctic sea-ice change tied to its mean state through thermodynamic processes. *Nature Climate Change*, 8(7), 599-603.

**Conclusion**

We greatly appreciate your insightful comments and valuable suggestions, which have provided important guidance for our subsequent revisions to the manuscript. We will carefully address all the concerns raised and further refine the work in line with the constructive feedback. We welcome any additional perspectives or advice as we proceed with the improvements.

Thank you for your time and efforts in reviewing our work.

**References**

- Aue, L., Vihma, T., Uotila, P., and Rinke, A.: New Insights Into Cyclone Impacts on Sea Ice in the Atlantic Sector of the Arctic Ocean in Winter, Geophys. Res. Lett., 49, e2022GL100051, https://doi.org/10.1029/2022GL100051, 2022.
- Belter, H. J., Krumpen, T., Von Albedyll, L., Alekseeva, T. A., Birnbaum, G., Frolov, S. V., Hendricks, S., Herber, A., Polyakov, I., Raphael, I., Ricker, R., Serovetnikov, S. S., Webster, M., and Haas, C.: Interannual variability in Transpolar Drift summer sea ice thickness and potential impact of Atlantification, The Cryosphere, 15, 2575–2591, https://doi.org/10.5194/tc-15-2575-2021, 2021.
- Bi, H., Liang, Y., and Chen, X.: Distinct Role of a Spring Atmospheric Circulation Mode in the Arctic Sea Ice Decline in Summer, J. Geophys. Res. Atmospheres, 128, e2022JD037477, https://doi.org/10.1029/2022JD037477, 2023.
- Cavallo, S. M., Frank, M. C., and Bitz, C. M.: Sea ice loss in association with Arctic cyclones, Commun. Earth Environ., 6, 44, https://doi.org/10.1038/s43247-025-02022-9, 2025.
- Crawford, A. D. and Serreze, M. C.: Does the Summer Arctic Frontal Zone Influence Arctic Ocean Cyclone Activity?, J. Clim., 29, 4977–4993, https://doi.org/10.1175/JCLI-D-15-0755.1, 2016.
- Ding, R., Huang, F., Shi, J., Zhao, C., and Xie, R.: Modulation of dominant thermodynamic processes and relay dynamic processes in Arctic sea ice rapid melting, Adv. Polar Sci., 41–50, https://doi.org/10.12429/j.advps.2024.0035, 2025.
- Docquier, D. and Koenigk, T.: A review of interactions between ocean heat transport and Arctic sea ice, Environ. Res. Lett., 16, 123002, https://doi.org/10.1088/1748-9326/ac30be, 2021.
- Finocchio, P. M., Doyle, J. D., Stern, D. P., and Fearon, M. G.: Short-term Impacts of Arctic Summer Cyclones on Sea Ice Extent in the Marginal Ice Zone, Geophys. Res. Lett., 47, e2020GL088338, https://doi.org/10.1029/2020GL088338, 2020.
- Finocchio, P. M., Doyle, J. D., and Stern, D. P.: Accelerated Sea Ice Loss from Late Summer Cyclones in the New Arctic, J. Clim., 35, 7751–7769, https://doi.org/10.1175/JCLI-D-22-0315.1, 2022.
- Holland, P. R. and Kimura, N.: Observed Concentration Budgets of Arctic and Antarctic Sea Ice, J. Clim., 29, 5241–5249, https://doi.org/10.1175/JCLI-D-16-0121.1, 2016.
- Holland, P. R. and Kwok, R.: Wind-driven trends in Antarctic sea-ice drift, Nat. Geosci., 5, 872–875, https://doi.org/10.1038/ngeo1627, 2012.
- Lukovich, J. V., Stroeve, J. C., Crawford, A., Hamilton, L., Tsamados, M., Heorton, H., and Massonnet, F.: Summer Extreme Cyclone Impacts on Arctic Sea Ice, J. Clim., 34, 4817–4834, https://doi.org/10.1175/JCLI-D-19-0925.1, 2021.
- Massonnet, F., Vancoppenolle, M., Goosse, H., Docquier, D., Fichefet, T., and Blanchard-Wrigglesworth, E.: Arctic sea-ice change tied to its mean state through thermodynamic processes, Nat. Clim. Change, 8, 599–603, https://doi.org/10.1038/s41558-018-0204-z, 2018.
- Mayer, M., Alonso Balmaseda, M., and Haimberger, L.: Unprecedented 2015/2016 Indo-Pacific Heat Transfer Speeds Up Tropical Pacific Heat Recharge, Geophys. Res. Lett., 45, 3274–3284, https://doi.org/10.1002/2018GL077106, 2018.
- Schroeter, S., Hobbs, W., Bindoff, N. L., Massom, R., and Matear, R.: Drivers of Antarctic Sea Ice Volume Change in CMIP5 Models, J. Geophys. Res. Oceans, 123, 7914–7938,

- https://doi.org/10.1029/2018JC014177, 2018.
- Stern, D. P., Doyle, J. D., Barton, N. P., Finocchio, P. M., Komaromi, W. A., and Metzger, E. J.: The Impact of an Intense Cyclone on Short-Term Sea Ice Loss in a Fully Coupled Atmosphere-Ocean-Ice Model, Geophys. Res. Lett., 47, e2019GL085580, https://doi.org/10.1029/2019GL085580, 2020.
- Sumata, H., De Steur, L., Gerland, S., Divine, D. V., and Pavlova, O.: Unprecedented decline of Arctic sea ice outflow in 2018, Nat. Commun., 13, 1747, https://doi.org/10.1038/s41467-022-29470-7, 2022.
- Sumata, H., De Steur, L., Divine, D. V., Granskog, M. A., and Gerland, S.: Regime shift in Arctic Ocean sea ice thickness, Nature, 615, 443–449, https://doi.org/10.1038/s41586-022-05686-x, 2023.
- Tian, Z., Liang, X., Zhang, J., Bi, H., Zhao, F., and Li, C.: Thermodynamical and Dynamical Impacts of an Intense Cyclone on Arctic Sea Ice, J. Geophys. Res. Oceans, 127, e2022JC018436, https://doi.org/10.1029/2022JC018436, 2022.
- Zhang, X., Walsh, J. E., Zhang, J., Bhatt, U. S., and Ikeda, M.: Climatology and Interannual Variability of Arctic Cyclone Activity: 1948–2002, J. Clim., 17, 2300–2317, https://doi.org/10.1175/1520-0442(2004)017%253C2300:CAIVOA%253E2.0.CO;2, 2004.

---

## Author Comment (AC2)

**Subject:**

Response to reviewer #2 for manuscript [egusphere-2025-3594: "Thermodynamic and dynamic drivers underlying extreme central Arctic sea ice loss"]

**Text in black: Reviewer's comments; Text in blue: Authors' responses.**

**General comments**

The study conducted by Li *et alii* investigates events of major sea ice concentration decrease in the Central Arctic. To do so, they use an Empirical Orthogonal Function decomposition of standardized sea ice concentration (SIC) anomalies, and use a set of complementary analyses, sea ice budget, temperature tendency analysis, sea ice dynamic decomposition and Rossby wave source identification, to determine the drivers of the two dominant modes of the SIC anomalies. The topic is of interest, though already extensively studied, and the study could contribute to the overall understanding of how sea ice in the Arctic evolves, in particular for the next few decades, during which the Central Arctic sea ice will become more and more vulnerable. Moreover, the large range of complementary analyses to determine the sources of sea ice loss are at first read enticing.

Unfortunately, the methods at the heart of those analyses and the data used to conduct them raise some serious concerns and I therefore have doubts as to whether they can support the claims from the authors. But more importantly, the study never addresses the potential role of the ocean to drive the ELSEs, while it is now considered as the first driver of sea ice loss. Moreover, it does not seem to disentangle the trend from the EOF modes, while it is likely hidden in the two dominant modes.

I do my best to detail my concerns below and to support them convincingly. I suspect addressing them properly will require a significant amount of work, including a total change of the methodology used. It should also lead to a significant change in the results.

**Major comments**

In a nutshell:
- Using reanalysis data to close the temperature and sea ice budget should not be done
- The temperature tendency equation does not include the heat flux coming from "below" (sea ice or ocean), while this can be the first order driver
- The ocean is never considered in the drivers of sea ice loss
- Sea ice concentration trends are not removed from the anomalies before conducting the EOF decomposition and are not discussed neither, which is a clear lack
- There might be some important between sea ice divergence and velocity divergence
- Many of the suggested causalities are not supported by the results and could be the other way around
- Many important methodological precisions are missing

Two major concerns are related to the methods and data used in the budgets (Methods sections 2.4 and 2.5 and Results sections 4 and 5).

Dear Reviewer,

We sincerely appreciate your thorough, insightful, and incisive comments on our manuscript, which were provided with the highest professional standards. We truly admire the meticulousness of your

review. Your comments have accurately identified key limitations of this study regarding methodological robustness and physical completeness. We also admire the logical clarity and well-organized format of your review comments.

All your comments have been carefully collated and are addressed point-by-point in the subsequent sections. In accordance with your suggestions, we will make substantial revisions and supplements to the research methodology and discussions. Your guidance has greatly benefited us and will significantly improve the quality of this manuscript.

Once again, we would like to extend our deepest gratitude for your diligent work and invaluable contributions.

**Reanalysis data fluxes should not be used to close a budget**

An important caveat of reanalysis products is that they are not physically consistent. Indeed, when assimilating observational data into the model state, spurious fluxes are introduced, breaking the conservation of some properties, including momentum and mass: in reanalysis, "the system state estimate can undergo jumps, implying implicit non-physical sources, and rendering very difficult the physical interpretation of the time-evolving state. Methods have been employed to smooth out the discontinuities over finite times, but still leaving artificial imbalances in the solution." (Wunsch & Heimbach, 2007). While reanalysis products provide the best estimate of the state of the climate, they should not be used to calculate budgets, as they cannot physically close them. A better, physically-consistent alternative to reanalysis would be State Estimates products, but those are costly to compute (e.g. ASTE, Nguyen et al., 2021) and typically not available for the kind of investigations conducted here.

Unfortunately, this study relies on reanalysis products, ERA5 and JRA55 to calculate a temperature budget, and PIOMAS to close a sea ice budget. This is a major issue, especially considering that the most important terms of the budgets (diabatic heating for the temperature; thermodynamics for the sea ice) are calculated by making the assumption that those budget are closed and that the residuals therefore correspond to the wanted term. Uncertainties are difficult to evaluate in reanalyses, and so it remains unknown whether using those data while significantly alter the results on the spatial and temporal scales considered here. But in doubts, I believe we have to make the assumption that the unphysical flux produced by data assimilation might not be negligible. Therefore, the method proposed here to evaluate the thermodynamical and dynamical contributions to sea ice loss is not sound. Note that this is a bit less worrisome for the sea ice budget, as the dynamical term is actually estimated by observations and not a reanalysis, but the thermodynamical term should still include not only the "real" thermodynamics but also a (hopefully small) spurious term related to the correction of the sea ice thickness by assimilation of observations into the PIOMAS $H_{eff}$.

Through your insightful comments and the valuable references you provided, we recognize that the non-physical fluxes introduced by data assimilation in reanalysis products indeed render them unsuitable for accurate budget calculations. This represents a theoretical limitation in our research methodology, and we fully concur with your concerns. However, as you noted, the more optimal physically consistent State Estimates products generally entail high computational costs. Moreover, for the spatiotemporal scales and variables targeted in our study, there are currently no publicly available, long-term alternative datasets. Under the present conditions, ERA5, JRA55, and PIOMAS remain widely recognized and extensively used in academia as robust data sources for analyzing large-scale climate and sea ice changes and their underlying mechanisms.

We followed the approach of previous studies to treat the residual term as the diabatic heating term (Yanai et al., 1992; Yao and Sun, 2016). Precisely because we recognized the limitation that this method is subject to certain systematic errors, we employed two independent datasets to maximize the reliability of our results.

For sea ice budget analysis, previous studies partitioned the SIC budget into advection, divergence, and a residual term that encompasses thermodynamic effects and mechanical processes associated with ridging and rafting (Holland and Kimura, 2016; Holland and Kwok, 2012). Using this method alongside observational data, they investigated the SIC budgets in both the Antarctic and Arctic. Schroeter et al. (2018) noted that replacing the SIC budget with the sea ice volume budget enables the incorporation of mechanical processes associated with ridging and rafting into SIT changes, allowing the residual term to directly represent thermodynamic components. This is exactly the method we choose and has now been widely adopted (Bi et al., 2023; Ding et al., 2025; Lukovich et al., 2021). Among these studies, Lukovich et al. (2021) conducted a detailed comparison of thermodynamic component contribution estimates from multiple studies, confirming the robustness of this method in treating thermodynamic components as the residual term.

In this study, observational data were prioritized wherever possible. For instance, SIM and SIC data were obtained from the NSIDC. Given that satellite platforms cannot directly observe SIT, and that no NSIDC datasets are suitable for the scope of this study, SIT data from the PIOMAS were employed instead. However, when conducting equation diagnostics by combining reanalysis and observational datasets, we must revisit the issue of budget closure. The Lagrange multiplier algorithm can address this issue to the greatest extent (Mayer et al., 2018). Using PIOMAS SIT data combined with NSIDC SIC data, Lukovich et al. (2021) investigated the impact of extreme cyclones on Arctic sea ice, while Ding et al. (2025) examined the relative contributions of thermodynamic and dynamic processes to Arctic sea ice decline. Both studies compared the original sea ice budget terms with the budget terms derived from the Lagrange multiplier algorithm and found nearly identical results, which confirms the reliability of using PIOMAS data combined with NSIDC data for sea ice budget diagnostics in this study.

The primary objective of this study is not to provide a fully accurate quantitative partitioning of thermodynamic and dynamic contributions, but rather to assess the differences in their relative importance and spatiotemporal distribution patterns within the framework of the most widely used datasets currently available. This approach aims to reveal clues about the physical mechanisms driving Arctic sea ice loss. We acknowledge that these data limitations inevitably introduce systematic errors. However, as numerous studies have shown, reanalysis data can still provide valuable mechanistic insights for analysis.

Your insightful comments have identified crucial avenues for improvement. We will explicitly discuss the range of uncertainties arising from this methodological limitation. Moreover, we will emphasize more strongly in the conclusions that our results should be interpreted as mechanistic explanations within the framework of specific datasets rather than precise physical quantifications. In the future, we will also actively track the release of physically consistent State Estimates products, with a view to validating or updating the conclusions of this study using more optimal datasets. We firmly maintain that despite this fundamental limitation, the exploration of the relative roles of dynamic and thermodynamic processes in this study can still provide valuable insights and a basis for discussion to enhance the understanding of Arctic sea ice variability mechanisms.

**The Temperature tendency does not account for the ocean or sea ice**

Equation (1) in section 2.4 equates the temperature tendency to the advection, the adiabatic heating and the diabatic. The tendency, the advection and the adiabatic heating are computed using the ERA5 reanalysis (see above for a major caveat of using this data for a budget). The last term, arguably the most important (I regret the authors did not show the comparison of all the terms), is estimated by considering that it is equal to the residuals of the budget. This could be true if 1. the reanalysis could be used to close the budget (I have argued above that it is not the case) and 2. if it was the only term missing. Unfortunately, I believe that the heat flux at the surface is not accounted for, in this equation. Indeed, sea ice or ocean are important heat sources or sinks and are therefore likely to provide an important heat flux. In equation (1), it is implicitly in the residual, but the text describing the equation makes me think that the authors are not aware of it: "The diabatic heating rate can also be directly estimated by summing the large-scale condensation heating rate, convective heating rate, vertical diffusion heating rate *[this would be the sensible heat flux between atmosphere and ocean/sea ice]*, solar radiation heating rate *[did the authors account for upward solar radiation proportional to the albedo of the surface? it seems not]*, and longwave radiation heating rate *[another heat flux for which sea ice or ocean need to be accounted for, but with no mention of it in the text]* based on the JRA-55 datasets." This is particularly worrisome as this study focuses on sea ice, but the equation is never used to link temperature tendency to sea ice! And in some cases (including during ELSEs), we can expect this flux to be the first order driver of the temperature tendency. Therefore this budget is not closed and, unless I missed something fundamental in the methods, what the authors consider as the diabatic heating is actually not the diabatic heating alone.

Because of those two major concerns, the results described in this study cannot be fully trusted. Many of those results are overall consistent with the scientific literature (e.g. the dominating importance of the thermodynamics over the dynamics in the sea ice budget, Le Guern-Lepage & Tremblay, 2023 or the importance of the "diabatic term" in the temperature tendency, over the other terms). But some other results are a bit at odds, to the best of my knowledge, e.g. the prominent importance of latent heat flux, which is rather supposed to be one or two orders of magnitude smaller than radiative and sensible heat fluxes (note that this could actually be related to another methodological issue, see last major comment).

We concur that estimating diabatic heating via the residual method using reanalysis datasets entails uncertainties, a common challenge in such studies within this field, as addressed above.

Furthermore, while the methodological description in our study is admittedly insufficiently detailed, we share full consensus with you regarding the physical connotation of diabatic heating. From the perspective of atmospheric column energy budget, surface heat fluxes (sensible heat, latent heat, and net radiation) represent the most critical lower-boundary source of atmospheric diabatic heating. Within our analytical framework, these heat fluxes from the ocean/sea ice surface are undoubtedly incorporated into the total diabatic heating term (i.e., the residual). This constitutes the fundamental physical rationale underpinning the linkage between atmospheric temperature variability and sea ice changes derived from our equation.

The textual description we provided of the diabatic heating components derived from JRA-55, which you cited, is merely intended to demonstrate an alternative estimation approach rather than to define the physical content of diabatic heating in the residual method we employed. Throughout all analyses in this study, the key "diabatic heating" term is consistently defined as the total budget

residual, which implicitly and fully incorporates the entire contribution of surface heat fluxes.

You suggested that we explicitly decompose the total diabatic heating into more detailed processes. While this represents a more stringent requirement, unambiguous and precise decomposition remains extremely challenging with current datasets, constrained by observational limitations and the accuracy of component outputs from reanalysis datasets. Our work focuses on evaluating the relative importance of total diabatic heating, which is inherently valuable and well aligned with the objectives of this study.

We thank you for prompting us to engage in more in-depth deliberations. To address your concerns, we will revise the manuscript as follows: (1) Clarify more explicitly the physical implication that the "diabatic heating" term in our methodology, as a total budget term, inherently incorporates surface heat fluxes; (2) Add a discussion section elaborating on the multiple processes encompassed by this integrated term and the challenges associated with its physical decomposition. The present study preliminarily establishes the critical role of thermodynamic processes in atmospheric temperature evolution against the backdrop of sea ice variability, using the total diabatic heating term as a comprehensive metric. Given that the diabatic heating term actually comprises multiple thermodynamic processes, we will supplement relevant discussions in the manuscript. Your comments also accurately identify the core task for future research: quantitatively isolating the direct contributions of individual components of diabatic heating. We will conduct in-depth studies on this topic in subsequent work, aiming to provide a more precise and mechanistic understanding of polar sea ice-climate interaction processes.

Finally, we will provide a detailed response to the issue concerning the magnitude of latent heat flux in the response to the seventh minor comment.

**What is the role of the ocean in the sea ice loss?**

This says it all. The ocean is a complete blind-spot of this study, while it now explains over half of the sea ice melt in the Central Arctic (e.g. Carmack et al., 2015, Oldenburg et al. 2024).

We fully concur with this critical perspective. Against the backdrop of global warming, the poleward oceanic heat transport has been exerting an increasingly pronounced influence on Arctic sea ice decline (Docquier and Koenigk, 2021). The budget diagnosis presented in Fig. 3 of this study indicates that thermodynamic contributions account for approximately 70% of the sea ice variations in the central Arctic region. Meanwhile, the high correlation coefficients (-0.77 and -0.91) between surface air temperature anomalies and sea ice variability modes underscore the significant influence of atmospheric processes (Fig. 4), yet fail to fully explain all thermodynamic contributions. This demonstrates that oceanic thermodynamic effects have emerged as an indispensable key factor. A prominent example is the widely discussed "Atlantification" process in recent years: the incursion of warm, saline water from the North Atlantic into the Arctic Ocean and the subsequent cascading climatic responses (Årthun et al., 2012; Polyakov et al., 2017).

Existing studies have demonstrated that Atlantification is steadily advancing eastward and poleward (Polyakov et al., 2017; Shu et al., 2021), and the latest observational evidence further indicates that its impacts have extended into the Makarov Basin (Polyakov et al., 2025). This signifies that sea ice changes in the central Arctic may have entered a new phase profoundly influenced by oceanic processes. Although this issue was recognized during the manuscript preparation, the complex, multi-scale nature of the mechanisms governing oceanic heat transport effects on sea ice precluded an in-depth investigation herein. This indeed represents a key limitation of the present work. Nevertheless, the diagnostic results of the thermodynamic contributions of EWSM and PASM in this study

incorporate all influencing factors and are therefore accurate.

It is worth noting that we have been conducting parallel studies focusing on the physical mechanisms by which Atlantification affects sea ice in the central Arctic. A manuscript of the relevant findings has been submitted to *Geophysical Research Letters* (Under Review). We will explicitly supplement the discussion section of this paper with an explanation of the limitations of the present study and highlight the significance of oceanic processes as a priority direction for future research. We believe that the analysis of atmospheric circulation impacts on sea ice presented in this paper still provides an important perspective for understanding Arctic sea ice variability, while the integration of oceanic processes will be the key to further advancing the current understanding.

**Are trends of sea ice concentration included in the EOF modes?**

In the preprocessing steps before decomposing the sea ice concentration into EOFs, the sea ice anomalies are computed by removing the climatology. But no trend seems to be removed. Considering the major changes that the sea ice is undergoing in the Arctic, I would expect the trend to be the dominant mode of the EOF decomposition. The authors first briefly mention that indeed the first mode of the non-normalized anomalies "spatially manifest as significant SIC anomaly signals along the Arctic marginal seas and the edge of the central Arctic" (l.220). The authors claims this is due to the summer signal; my guess is that this should also include the overall trend. The authors then normalize the anomalies to give equal weights to other seasons. But I would not expect this normalization to remove the trend. Yet, I am surprised to not see any mention of it when analysing the EOF decomposition of the standardized anomalies. Is that because it only appears in the third or higher order mode? Or because the method does indeed remove the trend? Or because the trend is actually not a major mode of variability? If the latter, this would be a major result that should be discussed. If not, I suspect it should be hidden somewhere and needs to be analysed and discussed. Moreover, in general, EOF decomposition studies tend to first remove the trend. I believe this needs to be done here as well. Note that this is not straight-forward for sea ice concentration, as this typically leads to sea ice concentrations above 1 at the beginning of the period of interest, and that a trend needs to be computed for each day-of-year (e.g. Richaud et al., 2025 for an example of day-of-year trend calculations for atmospheric variables).

The unstandardized EOF1 mode does contain a trend signal dominated by long-term external forcing, which indeed weakens after standardization. This means that the standardization to give equal weights to other seasons diminishes the relative advantage of long-term trend signals in terms of their variance contribution. Consequently, the relative weight of signals dominated by internal variability is enhanced. However, we also recognize that standardization is not a mathematically rigorous detrending method, and residual trend signals may still persist.

Accordingly, we will remove the linear trend from the daily time series at each grid point and directly compare the results before and after detrending to clearly illustrate the influence of the trend component.

We will include additional discussion on this important issue in the revised manuscript. We thank you for your constructive comments and the valuable reference you provided, which will be instrumental in helping us refine the technical aspects of our analysis.

**Sea ice divergence is not the same as Helmholtz divergent term**

In section 2.5, the sea ice dynamical term is decomposed into an advective and diverging term (eq. 3). Then in section 2.6, a Helmholtz decomposition of the (sea ice) velocity field is done, computing the divergent and rotational term. The text leads to think that the diverging term of eq. 3 and that of eq. 7 are equivalent. This is not the case and was (still is) very confusing to me. Those kinds of Helmholtz decompositions are typically done in rheological studies, but this is not the case here. Moreover, the text gives the impression that since dynamics can be decomposed into advection and divergence, and since the velocity field can be decomposed into divergence and vorticity, then the advection is equivalent to the vorticity: "By decomposing the standardized SIM fields into divergence (Fig. 9a and b) and vorticity (Fig. 9c and d) components, it can be observed that the sea ice advection primarily drives the anomalies in sea ice motion" (l. 312). This is obviously not the case. I do not understand why the advection was not directly computed, or if it was, why it wasn't shown and relied upon, rather than going through the rotational/vorticity. In any case, the Helmholtz decomposition does not bring anything to the study and I would suggest to drop it.

We sincerely appreciate your valuable comment and fully agree with your suggestion. In our original manuscript, we failed to clearly distinguish between the "dynamic terms of sea ice motion (advection and divergence terms)" and the "Helmholtz decomposition of the velocity field (divergent and rotational components)". This led to logical confusion. As you pointed out, these two concepts are not equivalent, and directly equating the advection effect with the rotational (vortical) component is physically inaccurate. Following your suggestion, we will remove all analyses related to the Helmholtz decomposition to avoid introducing unnecessary complexity. Instead, we will directly calculate and analyze the sea ice advection anomalies, and based on this, combine the analysis of divergence terms to more directly and clearly discuss the main driving mechanisms of sea ice motion anomalies.

**Many of the assertions are not supported by the analyses**

"The above analysis suggests that local surface temperature anomalies caused by local diabatic heating anomalies are the important factors in the formation of EWSM and PASM." (l.243): I do not see how this sentence is supported. It suggests that atmospheric (surface) temperatures drive the two EOF modes found in the study, on the basis that the spatial patterns of the composite temperature match the EOF mode patterns. But it could very well be (and I would guess likely is) the opposite, with the ice pattern driving the temperature. This is one example, amongst many, of a causal link claim made by the authors that could very well be the other way around. And that reversed causality is never explored or mentioned. Other examples include l. 285-286, l. 317-318, l.330-332, l.341-342, l.343-345 (list not exhaustive). Moreover, some other claims in the Conclusions and Discussions section do not seem to be really demonstrated in the paper: "Under thermodynamic dominance, both the EWSM and the PASM trigger water vapor and cloud feedbacks to sustain and enhance their development" (l.386-387). Section 4.2 does discuss this but does not provide any result to prove it and Figure 7 just gives some vague (not convincing to me) spatial coherence between the different metrics. Same with l.394: "the convergence and divergence of SIM exert a minor positive contribution to the EWSM": I could not find any substantial result in the main text that support this.

Proving the causal link is complex, requires using some causality methods (e.g. Liang-

Kleeman), and seems outside the scope of the study. Nonetheless, the claims of the authors are a bit too assertive to my opinion, and a more nuanced view on the direction of the links needs to be taken into account.

We acknowledge your insightful comment on a core issue in our manuscript: the overly definitive inference of causal directionality when explaining the complex interactions between sea ice and the atmosphere, and the lack of direct evidence from our analyses to support certain conclusions. We fully agree with your criticism, particularly regarding the arbitrary attribution of observed co-variations to a single directional forcing without rigorous causal testing, which is physically unjustified. Sea ice and the atmosphere form a tightly coupled system, and feedback processes between them are inherently bidirectional.

We will implement comprehensive revisions in the revised manuscript. We will thoroughly revise the entire text, changing all direct causal statements (e.g., "A drives B", "A triggers B") derived from spatial pattern similarity to more precise language describing "strong coupling", "co-variation", "concomitant occurrence", or "potential contribution to". We will add clarifications in the Discussion section explicitly stating that this study primarily reveals statistical correlations and spatial configurations of key variables in specific modes, which are consistent with known physical processes. Concurrently, we will candidly state that distinguishing causal directionality requires numerical experiments or specialized causal analysis methods, which lie beyond the scope of this observational diagnostic study and represent a direction for future work.

**Many important aspects of the methodology are missing**

The description of the methodology at the moment does not allow to reproduce the results. For example, the temperature tendency description (section 2.4) does not mention if this is for surface (2m) temperature, atmosphere-integrated temperature), boundary layer temperature or else. See also above for the lack of description of the trends of sea ice concentration, and other variables as well, if any trend is accounted for. The calculation of the climatology is not sufficiently described (see also minor comments on a suggestion to smooth it). None of the units of the terms are ever given (if they had been, it would have become obvious that the different "divergent" terms are not the same). The temporal threshold for the detection of ELSEs is not given. Looking at the figures, there might not be any, which then is a potential point of improvement of the study (see minor comment). Finally, references are missing for nearly all important equations used in the method.

The temperature tendency diagnosis uses surface temperature. We have addressed the issues of trends and climatology in our responses to specific comments. Units are omitted from the figures as they depict standardized variables. In fact, the spatial patterns of most variable anomalies are almost identical before and after standardization, and we will adjust the figure presentation accordingly. No temporal threshold is applied to ELSEs; they are identified simply when daily SIC falls below 1.5 SDs from the climatology. Several classic equations used in the methodology section indeed require citations, which we will add. All these deficiencies in the methodology section will be comprehensively supplemented in the revised manuscript.

**Minor comments**

- Many equations are not referenced, such as the temperature tendency equation, the Helmholtz decomposition, the diabatic heating rate calculation, etc. I know those are classic equations, but they can take alternative forms depending on the field of interest, and therefore a quick reference towards other papers using those equations in the same way would be relevant.

Thank you for your suggestions. We will carefully review the full text and add necessary references for key equations.

- Sea ice observation data: why only start in 1989? Sea ice concentration and motion data are available starting in 1979, which would give another decade of precious data on an else relatively short time series. This would give a more robust analysis.

You are absolutely correct. Available SIC data began in 1979. However, between 1979 and 1987, the area north of 84.5°N was subject to a satellite data Pole Hole Mask, which is too large for the central Arctic region studied in this paper. Furthermore, there is a major data gap in the period from December 1987 to January 1988. Considering these factors, the data selected for this study start from 1989.

The data user guide is available at https://nsidc.org/sites/default/files/documents/user-guide/nsidc-0051-v002-userguide.pdf

- Climatology calculation: the methods are not very explicit on the way the climatological mean is computed. It seems to be simply the mean of each day of the year, I suspect over the whole time series. A justified choice of the baseline would be good: 1979-2007 would avoid the recent decline period; or on the contrary only take the last 30 years to have the most recent behaviour, or the whole period? See Smith et al. (2025) for an in-depth discussion of why baseline are important. On top of that baseline aspect, it is conventional to smooth out the climatology when using daily data, by using a window around the considered day-of-year, yielding the advantage of increasing the sample size (see the MHW field of research, e.g. Hobday et al., 2016). This does not seem to have been the case in this study, when looking at Fig. 1.d).

We appreciate the valuable references you provided. We fully agree that defining a clear baseline period and applying climatological smoothing are essential to ensuring the robustness of the analytical results.

This study focuses on the sea ice variability in the central Arctic under the background of climate change. We therefore selected the period 1989–2022 as the overall climatological baseline. This encompasses both the relatively stable early phase and the recent period of rapid change. This selection facilitates an integrated analysis of sea ice responses to climate forcing. In addition, your observation is accurate. We did not perform smoothing on the daily climatology. We will provide a brief clarification and discussion of the baseline period selection and the treatment of climatological smoothing in the revised manuscript.

- ELSE definition: there does not seem to be any temporal threshold on the detection of the

ELSEs. In other word, a sea ice concentration below the 1.5 standard deviation threshold for 1 day would count as an ELSE, as would an event lasting a year. No discussion is brought on that aspect, and it seems to me that this requires some thinking and a different definition could yield different results. Considering the different temporal scales of the atmosphere, the ocean and the sea ice, the choice of the temporal threshold would give more or less weight on events that are likely to influence larger scale dynamics. I would recommend to filter out events shorter than a specific threshold, to be justified (e.g. 10 days? 1 month?). This would likely change the results in section 3.3: why is the 2008 event included as a significant one for the PASM mode, but not 2003?

We fully agree that distinguishing between transient anomalies and persistent events is critical in defining extreme events, as the latter are far more likely to exert a significant influence on large-scale climate dynamics. Our decision to adopt a definition without a duration threshold was motivated by the aim of performing an exploratory analysis. Specifically, this approach aims to capture all sea ice anomaly days below the statistical threshold initially and without omission, and to prevent the exclusion of potential signals from short-lived extremes due to an a priori persistence threshold.

- On the same aspect of ELSEs definition, I was very surprised to see that 2012 is not included in the list of ELSEs that are related to the EWSM and PASM modes, while it is the observational record of sea ice low. Considering its spatial pattern, I would expect it to be maybe in a PASM positive phase, but also with some EWSM negative contribution (wild, vaguely educated guess ;)). That does not seem to match Fig. 2, and I am curious as to why it is not in phase opposition with 2007 and why it is not more prominent. The study should at the very least discuss this aspect, considering the importance of 2012 in the recent sea ice evolution. Similarly, 2007 is also a very important sea ice low event: a couple of sentences on how it fits in the EOF modes story would be valuable.

The year 2012 is notable not only for its record minimum sea ice extent, but also for the super summer cyclone in the Pacific sector that has attracted widespread attention. Nevertheless, although the super cyclone induced a significant sea ice decline in the Pacific sector in 2012, this decline was concentrated primarily in the marginal regions of the Chukchi Sea and the East Siberian Sea, with a relatively limited impact on the central Arctic (Tian et al., 2022). This may be one of the reasons why the 2012 signal was not particularly prominent in our EOF analysis of the normalized sea ice anomaly field over the central Arctic.

We appreciate your valuable suggestion and will conduct a detailed examination of the sea ice decline in the central Arctic during 2007 and 2012, and discuss these variations in combination with the EWSM and PASM modes in the revised manuscript.

- The text only mentions the first two modes, which seem to explain 33% of the total variance. This means the other modes are also important. It seems that the authors would like to discuss the other modes in another study, but I think it would be important to at least mention how quickly the explained variance decreases with the other modes. Moreover, the fact that the first two modes explain "only" 33% of the total variance of sea ice concentration standardized anomalies seem to indicate that the EOF decomposition might not be the best

approach to explain the variability of sea ice. A discussion on this aspect seems important.

We will supplement a clarification of the decay in explained variance and a discussion of the applicability of the EOF method to this study in the revised manuscript (EOF3: 8.3%; EOF4: 5.4%). We preliminarily hypothesize that the relatively low cumulative explained variance is common in Arctic sea ice variability research, arising from the complex coupling and modulation of Arctic sea ice variability by atmospheric forcing, oceanic feedbacks, and sea ice dynamical processes.

- Figure S7 shows the composite differences of the standardized anomalies for the different heat fluxes. First, decomposing the radiation flux into solar (shortwave) and thermal (longwave) radiation would be valuable. Second, no description of how those composite differences and anomalies are computed, leaving the reader guessing the methodology. Is that the standardized anomalies of each flux? Or is it the composite anomalies of the fluxes but for the EOF modes of the standardized anomalies (of sea ice)? No units, no labels are given on the colorbar to help me decide. But I suspect it is the first option, looking at the colorbar values. If that's the case, it means each flux is normalized by its own standard deviation. But then we certainly cannot compare those fluxes together! The absolute value of the latent heat flux could be (and likely is, to the best of my knowledge) orders of magnitude smaller than the absolute value of the radiative fluxes: we have no information on that aspect in this manuscript. If that is the case, the claim that "Fig. S7 indicates that latent heat flux anomalies are prominent in the formation of the EWSM and PASM" (l.249-250) could be wrong, unfortunately.

First, we will further decompose the radiative flux into shortwave and longwave components. Second, as you have correctly inferred, Fig. S7 presents the standardized anomalies of each individual flux, which accounts for the absence of units on the colorbar. We acknowledge the flaw in our original description and conclusion. Given that the magnitude of the latent heat flux is substantially smaller than that of radiative fluxes, direct comparison of standardized variables is inappropriate for determining the dominant factor in the formation of the EWSM and PASM. We will revise this section accordingly and sincerely appreciate your insightful corrections.

- On a related topic, the reader misses critical information to understand how the SIM anomalies are standardized: are U&V standardized by the total velocity SD, or by U SD for U and by V SD for V? I suspect the first option, as the second would not make any sense and would prevent any comparison, but some weird patterns on Fig. 8 (e.g. on the Siberian Shelves for 8.b) cannot take off my mind that the second option might be used. Please provide the necessary details.

We adopted the first method. As you pointed out, the second method is methodologically flawed, specifically, as it would lead to erroneous sea ice drift directions. We will provide the necessary clarification in the revised manuscript.

- Divergence: "There is an out-of-phase relative variation of D between the Eastern and Western Hemispheres composited for EWSM mode" (l. 315). As already mentioned in the

major comments, I am very confused by the use of sea ice divergence in the sea ice (thickness) budget and in the Helmholtz decomposition; there is also a third "divergence" used in this study, defined by eq. 6 and written DF. This is not the same divergence as the other two, since it uses SIC, instead of Heff for the sea ice budget divergence and no sea ice for the Helmholtz decomposition. Yet, it does not seem to be distinguished in this section. I suspect that the DF and the ice thickness divergent term might be similar, but they would still show some differences.

We sincerely apologize for any confusion caused to you. First, we will remove all analyses related to Helmholtz decomposition to simplify the analytical framework (as stated in our response to Major Comment 5). Second, we will explicitly distinguish between the two remaining core concepts.

For the sea ice budget analysis, calculations were performed based on sea ice effective thickness. As noted in our response to the first question under Major Comment 1: "*Schroeter et al. (2018) noted that replacing the SIC budget with the sea ice volume budget enables the incorporation of mechanical processes associated with ridging and rafting into SIT changes, allowing the residual term to directly represent thermodynamic components.*" In contrast, the DF term is specifically intended to reveal the direct impact of advection processes on the leading modes of SIC variability. Thus, we opted for SIC-based calculations, which are not only computationally straightforward but also fully aligned with our analytical objectives. While minor discrepancies exist between results derived from sea ice effective thickness and SIC, the two approaches respectively address distinct yet complementary dimensions for interpreting the EOF modes, and do not affect the core conclusions of this study regarding the dominant physical processes governing the modes. We will add clarifications to this effect in the revised manuscript.

- The whole section 6 on the Rossby wave source and teleconnection seems out of place. It does not use the same approach (why not use the composite differences as done for all other aspects?), it is not clearly linked to the EOF modes, Fig. 11 only shows the Pacific side of the Arctic (why not the rest) and its contribution to new scientific knowledge is not obvious to me (I believe the Pacific role in the Rossby wave generation is well known and that the spatial pattern of the T-N wave activity flux has also already been documented extensively). Therefore, I do not understand the contribution of this section to the general scientific knowledge. But I admit that this is a bit far from my domain of expertise and I might simply not be knowledgeable enough to understand its significance. I let the other reviewer(s), the editor and the authors judge on that aspect.

First, we chose the continuous case from 2002 for demonstration, primarily considering the prominent temporal propagation characteristics of waves. In fact, the results derived from the composite difference method are nearly identical to those in Fig. 11, which further validates the representativeness of the selected typical case.

Second, the core objective of Section 6 is to dynamically link the sea ice anomaly modes over the central Arctic identified in this study to the source regions and propagation pathways of mid-latitude waves. This work thus builds a bridge between the two research perspectives of "local sea ice-atmosphere interactions" and "mid-latitude teleconnections", which carries certain implications for future research.

Finally, we will revise the spatial scope of Fig. 11 according to your suggestion.

- Oceanic discussion (l. 413-435): This is the first and only time that the ocean is considered in the story. Unfortunately, the Sverdrup balance is not valid for the Arctic (e.g. Timmermans & Marshall, 2020), and the inflow of warm Atlantic water is governed 8 by a complex set of atmospheric and oceanic interactions. Moreover, the Sverdrup balance is brought up in an attempt to explain discrepancies between Fig. 5b and S10e. The issue is that those are not showing the same thing at all! Fig. S10e shows surface temperature anomalies (in K) while Fig. 5b shows the diabatic rate (which should be in K/s); comparing that latter figure to the temperature tendency anomalies could be done, but not to the actual temperature anomalies. Hence, we should not expect a similarity between Fig 5b and S10e. Regarding the oceanic heat transport in the Arctic, a good source of information is Docquier & Koenigk (2021) and Docquier et al. (2021). Check also Polyakov et al. (2023). Many other papers investigate the role of heat inflow into the Arctic and would be a much more reliable and convincing source of explanation of Atlantification than the proposed hand-wavy Sverdrup balance that is not applicable.

We acknowledge that the discussion in this section lacks sufficient scientific justification. The north-south gradient of the Coriolis parameter is excessively small in the Arctic, rendering the traditional Sverdrup balance inapplicable. We have conducted a preliminary investigation into the impacts of Arctic Atlantification on sea ice in the central Arctic (as noted in our response to the third issue under Major Comments). We will carefully revise the discussion in this section.

**Suggestions, technical details and typos**

**Text**

- Use of "vorticity" (e.g. l. 313), "rotational" (e.g. l. 154) and "nondivergent" (e.g. Fig. 9) names for the same term: please pick one term and stick to it. Same for divergent vs. irrotational.
- l.10: remove "First,"
- l.14: "which highlight common characteristics of sea ice variations." We could argue that this is not really the case, and that concentration anomalies are simply a convenient metric to observe, but that ice thickness would be much better to really understand sea ice variation.
- l. 28, "approximately twice the global average": the most recent estimates rather suggest 3 to 4 times (e.g. Rantanen et al. 2022)
- l. 33, "This rapid warming in the Arctic has led to a significant decline trend in Arctic sea ice ": one could argue that the decline in Arctic sea ice has led to the rapid warming, more than the opposite (albedo positive feedback). Please nuance this sentence.
- l. 47: Voosen (2020) seems like a journalistic piece, not a scientific article. While it is an interesting one, please provide a peer-reviewed paper. Moreover, I could not find any part of that (short) piece that talks about a dynamic and thermodynamic coupling... was that LLM-generated?
- l. 51: "Arctic sea ice concentration" or "extent"
- l. 54: Sticker et al. (2025) would be a good, recent addition here

- l. 61: Hoffman et al. (2025) would be a good, recent addition here
- l.63, "the reduction of Arctic sea ice is spatially heterogeneous, which is attributable to the spatial variation of thermodynamic and dynamic processes driven by atmospheric and oceanic circulation. [...] The trend of sea ice reduction is notably significant in the marginal seas along the Eurasian and the North American coast, while the sea ice from north of Greenland and Canadian Arctic Archipelago to the pole remains relatively stable". I agree that atmospheric and oceanic circulation play an important role in generating spatial variability, but not those described by the text. The difference between shelves and northern part is simply due to astronomical considerations (less solar radiation close to the pole than further south, e.g. Maksym 2019) ... Please rephrase.
- l.71, "the perennial sea ice in the central Arctic has begun to undergo extreme reductions in recent years": "recent" is subjective, but it has been a few decades by now, so I would remove the "begun [...] in recent years"
- l.76, "This is because the sea ice in the central Arctic region is predominantly multi-year thick ice, and the absolute value of sea ice variation in winter and spring is relatively small". It is also (and maybe primarily) because the winter and spring sea ice extent is strongly geographically constrained by the surrounding continents, and that there is therefore less degree of freedom (e.g. Maksym, 2019). Please rephrase.
- l. 179-186: This is a good introduction, though maybe a bit too detailed. You could shorten to only highlight the relevant definition.
- l.187, "climatological SIC": is that annual mean? seasonal? day-of-year? Maybe worth considering a time-varying definition. See also minor comment for smoothing suggestion, to have a more statistically robust definition. If it is annual (as Fig 1 seems to suggest), it needs some discussion, as it will induce a significant ELSE detection bias between winter and summer.
- l.193, "which validates the rationality of the definition of the central Arctic in this study": except for JAS, for which the probability of SIC>90 % represents a small fraction of the Central Arctic. See above suggestion to use a time-varying definition.
- l.225-226: please provide references for those EWSM and PASM: are those names yours? Or do they come from other studies? Do they match other studies?
- l. 241: how is the spatial correlation computed? Why is it only computed for air temperature and not for the diabatic heating, heat fluxes, etc? Also, see major comments: correlation is not causation.
- l. 354, "Considering that the most pronounced EWSM signal occurs during the winter of 2002 [...]"
- l. 454-455, "The unpresented EOF3 and EOF4 in this study are primarily related to anomalous temperature advection from mid-latitudes, with significant increases in dynamic contributions.": I don't really understand the distinction made between dynamic and thermodynamic, then. To me, the advection of heat would lead to a change in thermodynamics, not dynamics, which would rather be controlled by momentum fluxes, not heat fluxes. So this sentence seems self-contracting to me.

**Figures**

- All: please provide labels + units on all colorbars and don't hesitate to also add title above the different panels to make sure the reader can quickly understand what they are looking at.

Try to be consistent and homogeneous between figures, keeping the same latitude boundaries, projections, row/column orientation, etc.

- Fig. 1: The colormap for panels a and b is not sequential, consider using another one; please mask regions where there is no ice, instead of plotting the 0% SIC. Panel e (and left panels in figure 2) are great! I would suggest making those a bit wider to see better.
- Fig. 3: this figure is a bit confusing because at first glance, it is not clear that adding advection and divergence leads to the Dynamics term. Please consider stacking them to reducing their width and adding transparency to make it more obvious that there are only two terms and that you decompose one of them into two.
- Figures 6 and 9: I would suggest to transpose the figures, putting the EWSM and PASM as rows instead of columns, to match the other figures (e.g. Fig 7 and those in SI). Keeping the same convention would allow the reader to be able to skim through and compare the figures in a more intuitive way.
- Figs. 10 and 11: Why change the projection? Other figures use a North Polar Stereo projection while 10 and 11 are cylindrical (?).
- Fig. 11: Why not plotting the whole hemisphere? At least for a) and d).
- Fig. 12: Great schematic! Not sure it is very colorblind-friendly, but I like it. I am unfortunately not convinced by the content, because of all the reasons detailed in the major comments...

We sincerely appreciate your meticulous review and careful revisions of our manuscript. We have carefully considered all your suggestions and incorporated them into the revised version, which will significantly enhance the clarity, accuracy, and overall quality of our work.

**References**

- Carmack, E., Polyakov, I., Padman, L., Fer, I., Hunke, E., Hutchings, J., Jackson, J., Kelley, D., Kwok, R., Layton, C., Melling, H., Perovich, D., Persson, O., Ruddick, B., Timmermans, M.-L., Toole, J., Ross, T., Vavrus, S., and Winsor, P.: Toward Quantifying the Increasing Role of Oceanic Heat in Sea Ice Loss in the New Arctic, Bulletin of the American Meteorological Society, 96, 2079–2105, https://doi.org/ 10.1175/BAMS-D-13-00177.1, 2015.
- Docquier, D. and Koenigk, T.: A review of interactions between ocean heat transport and Arctic sea ice, Environ. Res. Lett., 16, 123002, https://doi.org/10.1088/1748 9326/ac30be, 2021.
- Docquier, D., Koenigk, T., Fuentes-Franco, R., Karami, M. P., and Ruprich-Robert, Y.: Impact of ocean heat transport on the Arctic sea-ice decline: a model study with EC Earth3, Clim Dyn, 56, 1407–1432, https://doi.org/10.1007/s00382-020-05540-8, 2021.
- Hobday, A. J., Alexander, L. V., Perkins, S. E., Smale, D. A., Straub, S. C., Oliver, E. C. J., Benthuysen, J. A., Burrows, M. T., Donat, M. G., Feng, M., Holbrook, N. J., Moore, P. J., Scannell, H. A., Sen Gupta, A., and Wernberg, T.: A hierarchical approach to defining marine heatwaves, Progress in Oceanography, 141, 227–238, https://doi.org/10.1016/j.pocean.2015.12.014, 2016.
- Hoffman, L., Massonnet, F., and Sticker, A.: Probabilistic Forecasts of September Arc tic Sea Ice Extent at the Interannual Timescale With Data-Driven Statistical Models, Journal of Geophysical Research: Machine Learning and Computation, 2, e2025JH000669,

https://doi.org/10.1029/2025JH000669, 2025.

- Le Guern-Lepage, A. and Tremblay, B. L.: Disentangling Dynamic from Thermodynamic Summer Ice Area Loss from Observations (1979–2021): A Potential Mechanism for a "First-Time" Ice-Free Arctic, Journal of Climate, 36, 7693–7713, https: //doi.org/10.1175/JCLI-D-22-0628.1, 2023.

- Maksym, T.: Arctic and Antarctic Sea Ice Change: Contrasts, Commonalities, and Causes, Annual Review of Marine Science, 11, 187–213, https://doi.org/10.1146/ annurev-marine-010816-060610, 2019.

- Nguyen, A. T., Pillar, H., Ocaña, V., Bigdeli, A., Smith, T. A., & Heimbach, P. (2021). The Arctic Subpolar gyre sTate Estimate: Description and assessment of a data-constrained, dynamically consistent ocean-sea ice estimate for 2002–2017. Journal of Advances in Modeling Earth Systems, 13, e2020MS002398. https://doi.org/10. 1029/2020MS002398

- Oldenburg, D., Kwon, Y.-O., Frankignoul, C., Danabasoglu, G., Yeager, S., and Kim, W. M.: The Respective Roles of Ocean Heat Transport and Surface Heat Fluxes in Driving Arctic Ocean Warming and Sea Ice Decline, Journal of Climate, 37, 1431–1448, https://doi.org/10.1175/JCLI-D-23-0399.1, 2024.

- Polyakov, I. V., Ingvaldsen, R. B., Pnyushkov, A. V., Bhatt, U. S., Francis, J. A., Janout, M., Kwok, R., and Skagseth, Ø.: Fluctuating Atlantic inflows modulate Arctic atlantification, Science, 381, 972–979, https://doi.org/10.1126/science.adh5158, 2023.

- Richaud, B., Dowd, M., Renkl, C., and Oliver, E. C. J.: Sea Ice Nonlinearities Act to Rectify and Filter Oceanic and Atmospheric Forcing, Journal of Climate, 38, 4573–4588 https://doi.org/10.1175/JCLI-D-24-0485.1, 2025.

- Smith, K. E., Sen Gupta, A., Amaya, D., Benthuysen, J. A., Burrows, M. T., Capotondi, A., Filbee-Dexter, K., Frölicher, T. L., Hobday, A. J., Holbrook, N. J., Malan, N., Moore, P. J., Oliver, E. C. J., Richaud, B., Salcedo-Castro, J., Smale, D. A., Thomsen, M., and Wernberg, T.: Baseline matters: Challenges and implications of different marine heatwave baselines, Progress in Oceanography, 231, 103404, https: //doi.org/10.1016/j.pocean.2024.103404, 2025.

- Sticker, A., Massonnet, F., Fichefet, T., DeRepentigny, P., Jahn, A., Docquier, D., Wyburn-Powell, C., Quint, D., Shivers, E., and Ortiz, M.: Seasonality and scenario dependence of rapid Arctic sea ice loss events in CMIP6 simulations, The Cryosphere, 19, 3259–3277, https://doi.org/10.5194/tc-19-3259-2025, 2025.

- Timmermans, M.-L. and Marshall, J.: Understanding Arctic Ocean Circulation: A Review of Ocean Dynamics in a Changing Climate, Journal of Geophysical Research: Oceans, 125, e2018JC014378, https://doi.org/10.1029/2018JC014378, 2020.

- Wunsch, C. and Heimbach, P.: Practical global oceanic state estimation, Physica D, 230, 197–208, 2007.

**References**

Årthun, M., Eldevik, T., Smedsrud, L. H., Skagseth, Ø., and Ingvaldsen, R. B.: Quantifying the Influence of Atlantic Heat on Barents Sea Ice Variability and Retreat*, J. Clim., 25, 4736–4743, https://doi.org/10.1175/JCLI-D-11-00466.1, 2012.

Bi, H., Liang, Y., and Chen, X.: Distinct Role of a Spring Atmospheric Circulation Mode in the Arctic

Sea Ice Decline in Summer, J. Geophys. Res. Atmospheres, 128, e2022JD037477, https://doi.org/10.1029/2022JD037477, 2023.

Ding, R., Huang, F., Shi, J., Zhao, C., and Xie, R.: Modulation of dominant thermodynamic processes and relay dynamic processes in Arctic sea ice rapid melting, Adv. Polar Sci., 41–50, https://doi.org/10.12429/j.advps.2024.0035, 2025.

Docquier, D. and Koenigk, T.: A review of interactions between ocean heat transport and Arctic sea ice, Environ. Res. Lett., 16, 123002, https://doi.org/10.1088/1748-9326/ac30be, 2021.

Holland, P. R. and Kimura, N.: Observed Concentration Budgets of Arctic and Antarctic Sea Ice, J. Clim., 29, 5241–5249, https://doi.org/10.1175/JCLI-D-16-0121.1, 2016.

Holland, P. R. and Kwok, R.: Wind-driven trends in Antarctic sea-ice drift, Nat. Geosci., 5, 872–875, https://doi.org/10.1038/ngeo1627, 2012.

Lukovich, J. V., Stroeve, J. C., Crawford, A., Hamilton, L., Tsamados, M., Heorton, H., and Massonnet, F.: Summer Extreme Cyclone Impacts on Arctic Sea Ice, J. Clim., 34, 4817–4834, https://doi.org/10.1175/JCLI-D-19-0925.1, 2021.

Mayer, M., Alonso Balmaseda, M., and Haimberger, L.: Unprecedented 2015/2016 Indo-Pacific Heat Transfer Speeds Up Tropical Pacific Heat Recharge, Geophys. Res. Lett., 45, 3274–3284, https://doi.org/10.1002/2018GL077106, 2018.

Polyakov, I. V., Pnyushkov, A. V., Alkire, M. B., Ashik, I. M., Baumann, T. M., Carmack, E. C., Goszczko, I., Guthrie, J., Ivanov, V. V., Kanzow, T., Krishfield, R., Kwok, R., Sundfjord, A., Morison, J., Rember, R., and Yulin, A.: Greater role for Atlantic inflows on sea-ice loss in the Eurasian Basin of the Arctic Ocean, Science, 356, 285–291, https://doi.org/10.1126/science.aai8204, 2017.

Polyakov, I. V., Pnyushkov, A. V., Charette, M., Cho, K.-H., Jung, J., Kipp, L., Muilwijk, M., Whitmore, L., Yang, E. J., and Yoo, J.: Atlantification advances into the Amerasian Basin of the Arctic Ocean, Sci. Adv., 11, eadq7580, https://doi.org/10.1126/sciadv.adq7580, 2025.

Schroeter, S., Hobbs, W., Bindoff, N. L., Massom, R., and Matear, R.: Drivers of Antarctic Sea Ice Volume Change in CMIP5 Models, J. Geophys. Res. Oceans, 123, 7914–7938, https://doi.org/10.1029/2018JC014177, 2018.

Shu, Q., Wang, Q., Song, Z., and Qiao, F.: The poleward enhanced Arctic Ocean cooling machine in a warming climate, Nat. Commun., 12, 2966, https://doi.org/10.1038/s41467-021-23321-7, 2021.

Tian, Z., Liang, X., Zhang, J., Bi, H., Zhao, F., and Li, C.: Thermodynamical and Dynamical Impacts of an Intense Cyclone on Arctic Sea Ice, J. Geophys. Res. Oceans, 127, e2022JC018436, https://doi.org/10.1029/2022JC018436, 2022.

Yanai, M., Li, C., and Song, Z.: Seasonal Heating of the Tibetan Plateau and Its Effects on the Evolution of the Asian Summer Monsoon, J. Meteorol. Soc. Jpn. Ser II, 70, 319–351, https://doi.org/10.2151/jmsj1965.70.1B_319, 1992.

Yao, X. and Sun, J.: Thermal forcing impacts of the easterly vortex on the east-west shift of the subtropical anticyclone over Western Pacific Ocean, J. Trop. Meteorol., 22, 51–56, https://doi.org/10.16555/j.1006-8775.2016.01.006, 2016.